# ATTENTION SINKS AND COMPRESSION VALLEYS IN LLMS ARE TWO SIDES OF THE SAME COIN

**Enrique Queipo-de-Llano**[*,1], **Álvaro Arroyo**[*,1], **Federico Barbero**[1],
**Xiaowen Dong**[1], **Michael Bronstein**[1,2], **Yann LeCun**[3], **Ravid Shwartz-Ziv**[3]
[1]University of Oxford [2]AITHYRA [3]New York University

## ABSTRACT

Attention sinks and compression valleys have attracted significant attention as two puzzling phenomena in large language models, but have been studied in isolation. In this work, we present a surprising connection between attention sinks and compression valleys, tracing both to the formation of massive activations in the residual stream. We prove theoretically that massive activations necessarily produce representational compression and establish bounds on the resulting entropy reduction. Through experiments across several models (410M–120B parameters), we confirm that when the beginning-of-sequence token develops extreme activation norms in the middle layers, both compression valleys and attention sinks emerge simultaneously. Targeted ablation studies validate our theoretical predictions. This unified view motivates us to propose the *Mix-Compress-Refine* theory of information flow, as an attempt to explain how LLMs organize their computation in depth by controlling attention and representational compression via massive activations. Specifically, we posit that Transformer-based LLMs process tokens in three distinct phases: (1) broad mixing in the early layers, (2) compressed computation with limited mixing in the middle layers, and (3) selective refinement in the late layers. Our framework helps explain why embedding tasks perform best at intermediate layers, whereas generation tasks benefit from full-depth processing, clarifying differences in task-dependent representations.

## 1 INTRODUCTION

Large Language Models (LLMs) have become remarkably capable, yet how they process information through their layers remains poorly understood. Two phenomena have particularly puzzled researchers: *attention sinks*, where attention heads mysteriously collapse their focus onto semantically uninformative tokens (Xiao et al., 2024), and *compression valleys*, where intermediate representations show unexpectedly low entropy despite the model's high-dimensional space (Skean et al., 2025).

These phenomena appear paradoxical: why would powerful models waste attention on meaningless tokens, and why would representations compress in the middle of processing? Previous work has explained attention sinks through positional biases (Gu et al., 2025) and over-mixing prevention (Barbero et al., 2025a), while compression valleys have been explained through an information bottleneck theory (Skean et al., 2025). However, the precise reasons why they emerge remain unclear and no formal link has been established between them.

We reveal that attention sinks and compression valleys are two manifestations of a single mechanism: *massive activations* in the residual stream. These extremely large-magnitude features emerge on specific tokens (typically the beginning-of-sequence token, BOS), create both effects simultaneously, and act as input-agnostic biases. While prior work linked massive activations to attention sinks (Sun et al., 2024; Cancedda, 2024), we prove they also drive compression: when a single token's norm dominates others, it necessarily creates a dominant singular value in the representation matrix, explaining the compression. Experiments across several models (410M–120B parameters) confirm this unified mechanism, connecting both phenomena via the massive activations.

---

[*]Denotes equal first authorship. Correspondence to alvaro.arroyo@eng.ox.ac.uk and
enrique.queipodellanoburgos@reuben.ox.ac.uk

This unified mechanism reveals how transformers organize computation across depth through the *Mix-Compress-Refine* framework: massive activations control three distinct processing phases. Early layers (0–20% depth) mix information broadly via diffuse attention. Middle layers (20–85%) compress representations while halting mixing through attention sinks, both triggered by massive activation emergence. Late layers (85–100%) selectively refine through localized attention as norms equalize. This phase structure explains task-specific optimal depths: embedding tasks peak during compression, while generation requires full refinement.

Our contributions are:

- We empirically demonstrate that attention sinks and compression valleys emerge simultaneously in middle layers across several models (410M–120B parameters).
- We prove that massive activations mathematically require compression, providing tight bounds on entropy reduction and singular value dominance (Theorem 1).
- We validate causality through targeted ablations: removing massive activations eliminates both compression and reduces attention sinks.
- We propose the Mix-Compress-Refine theory of information flow, explaining how transformers organize computation into three distinct phases.
- We show this framework helps resolve the puzzle of task-dependent optimal depths: embedding tasks peak during compression while generation requires full refinement.

## 2 BACKGROUND AND RELATED WORK

In this paper, we study decoder-only transformers with $L$ layers, hidden dimension $d$, and $H$ attention heads per layer. For a sequence of $T$ tokens, let $\mathbf{x}_i^{(\ell)} \in \mathbb{R}^d$ denote token $i$'s representation at layer $\ell$, and $\mathbf{X}^{(\ell)} \in \mathbb{R}^{T \times d}$ the full representation matrix. Attention weights $\alpha_{ij}^{(\ell,h)}$ from token $i$ to $j$ in head $h$ satisfy causal masking ($\alpha_{ij} = 0$ for $j > i$). Full architecture provided in Appendix A.1.

**Key Metrics.** For a representation matrix $\mathbf{X}$ with singular values $\sigma_1 \geq \sigma_2 \geq \ldots \geq \sigma_r$, we measure compression via the **matrix-based entropy**:

$$H(\mathbf{X}) = -\sum_{j=1}^{r} p_j \log p_j, \quad \text{where } p_j = \sigma_j^2 / \|\mathbf{X}\|_F^2 \tag{1}$$

Low entropy indicates compression into few dominant directions. The **anisotropy** $p_1 = \sigma_1^2 / \|\mathbf{X}\|_F^2$ measures directional bias (Razzhigaev et al., 2023) (near 1 = extreme bias, near $1/r$ = isotropy). For token position $k$, the **attention sink score** and **sink rate** (Gu et al., 2025) are:

$$\text{sink-score}_k^{(\ell,h)} = \frac{1}{T} \sum_{t=0}^{T-1} \alpha_{tk}^{(\ell,h)}, \quad \text{sink-rate}_k^{(\ell)} = \frac{1}{H} \sum_{h=1}^{H} \mathbb{I}\left(\text{sink-score}_k^{(\ell,h)} \geq \tau\right) \tag{2}$$

with threshold $\tau = 0.3$ (unless otherwise stated), and $\mathbb{I}$ denotes the indicator function. We focus on the BOS token, the primary sink across models.

**Attention Sinks.** Attention heads mysteriously focus on semantically uninformative tokens (e.g., BOS) across diverse models and scales (Xiao et al., 2024). While Barbero et al. (2025a) argues they prevent over-mixing, Cancedda (2024) relates them to spectral subspaces, and Gu et al. (2025) traces emergence to pretraining, no work has yet examined their depth-wise organization.

**Compression Valleys.** Transformer representations compress dramatically in middle layers, where the matrix-based entropy drops significantly before recovering (Skean et al., 2025). This universal pattern coincides with increased anisotropy ($p_1 > 0.9$) (Razzhigaev et al., 2023) and near-linear layer transitions (Razzhigaev et al., 2024). Paradoxically, these compressed representations excel at embedding tasks despite their reduced dimensionality. The mechanism remained unknown, with only information bottleneck hypotheses lacking causal evidence (Skean et al., 2025).

**Massive Activations.** Sun et al. (2024) identified extremely large-magnitude features in transformer residual streams, with individual neurons exceeding typical activations by factors of $10^3$–$10^6$. These "massive activations" consistently appear on delimiter and special tokens (particularly

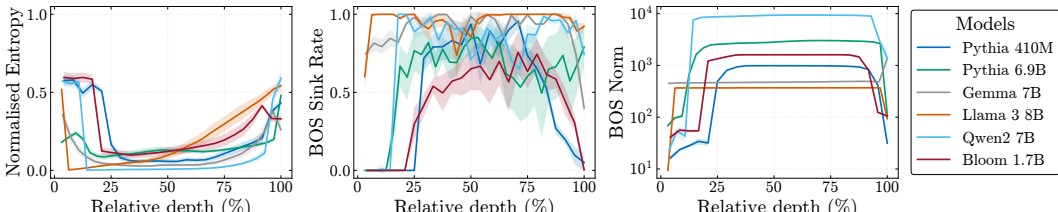

Figure 1: **Attention sinks and compression valleys emerge simultaneously when BOS tokens develop massive activations.** Normalized entropy (**left**), BOS sink rate (**middle**), and BOS token norm (**right**) across layers for six models evaluated on GSM8K. All three phenomena align precisely: when BOS norms spike by factors of $10^3$–$10^4$ (right panel), entropy drops below 0.5 bits (left) and sink rates surge to near 1.0 (middle), confirming our unified mechanism hypothesis.

the BOS token), acting as input-agnostic biases. Furthermore, Sun et al. (2024) found a link between the emergence of massive activations and attention sinks, which was reinforced in Barbero et al. (2025b); Yona et al. (2025). *However, none of these works link this phenomenon to representational structure or a unified theory of information flow in LLMs.*

**Computation Across Depth in Transformers.**  Several works have sought to understand the evolution of representations in Transformer-based models from a theoretical perspective. Dong et al. (2021) proved that the repeated application of self-attention leads to rank collapse in simplified settings without residual connections. Geshkovski et al. (2023) analyze self-attention dynamics and show tokens cluster over depth. Wu et al. (2024) studied how layer normalization and attention masks affect information propagation, finding that normalization can prevent rank decay. Other empirical work examines intermediate layer outputs directly. We highlight the LogitLens (Nostalgebraist, 2020) and TunedLens (Belrose et al., 2023), which decode hidden states using the model unembedding matrix and an affine probe per layer, respectively. Furthermore, Lad et al. (2024) measures the sensitivity to delete and swap interventions across layers and argues at different stages of depth-wise inference. Most recently, Csordás et al. (2025) argue that deeper LLMs underutilize their additional layers, with later layers mainly refining probability distributions rather than composing novel computations. While these studies illuminate layerwise behavior, none provides a unified mechanism that explains *why* stages form in depth or predicts *when* they should appear.

**The Gap This Work Addresses.**  Thus far, attention sinks have been tied to massive activations while compression has remained a separate observation without a causal mechanism. In this work, we document the synchronized dynamics of these phenomena, showing that the same massive activations that create sinks are also the main driver of compression. Building on their co-emergence, we propose a three-stage theory in which residual-stream norms simultaneously regulate mixing in attention heads and compression in representations. Finally, we connect the mechanism to downstream behavior, distinguishing between embedding-style and generation tasks.

# 3  MASSIVE ACTIVATIONS DRIVE BOTH ATTENTION SINKS AND COMPRESSION

> **Key Insight:** Attention sinks and compression valleys are not separate phenomena but two consequences of massive activations in the residual stream. We prove theoretically that when BOS token norms exceed others, they necessarily create a dominant singular value, causing compression and coinciding with attention sinks. This unification reveals that a single mechanism controls both representation structure and attention in middle layers.

## 3.1  EMPIRICAL CORRELATION: SYNCHRONIZED EMERGENCE ACROSS MODELS

We first empirically document that attention sinks and compression valleys emerge simultaneously across model families and scales. Figure 1 shows the layer-wise evolution of three metrics across six models (Pythia 410M/6.9B, LLaMA3 8B, Qwen2 7B, Gemma 7B, Bloom 1.7B): (1) matrix-based entropy $H(\mathbf{X}^{(\ell)})$, (2) sink-rate$_0^{(\ell)}$, and (3) BOS token norm $\|\mathbf{x}_0^{(\ell)}\|$. We compute these metrics for all 7.5K training examples in GSM8K (Cobbe et al., 2021), plotting the mean and standard deviation at each layer.

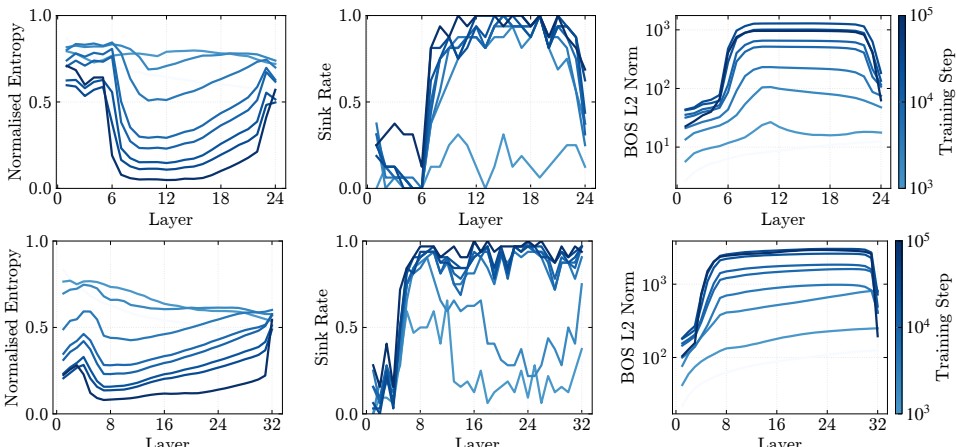

Figure 2: **The coupled emergence of massive activations, compression, and sinks develops early in training and persists.** Evolution of normalized entropy **(left)**, sink rate **(middle)**, and BOS norm **(right)** across training checkpoints (1–143k steps) for Pythia 410M, 6.9B. All three phenomena emerge together around step 1k and remain synchronized throughout training, indicating this organization is learned early.

We observe that all **three patterns align precisely**. When the BOS norm spikes to factors of $10^3$–$10^4$ (typically layers 0–5 depending on model depth), entropy simultaneously drops and sink rates surge. We compute the Pearson correlation between the change in BOS norm and entropy, obtaining $r = -0.9 \pm 0.18$ across models, while BOS norm and sink rate correlate at $r = 0.58 \pm 0.25$[1].

We highlight that this synchronization is remarkably consistent. While sink rates vary with prompt content, the *layer index* where these phenomena emerge is fixed for each model, and the massive activation is deterministic. For instance, in Pythia 410M, the transition consistently occurs at layer 5 regardless of input, suggesting an architectural rather than input-dependent mechanism. We show how these transitions emerge during training in Figure 2. We point the reader to Appendix B.1 for details on experiments, larger models, and a note on GPT OSS (Agarwal et al., 2025).

### 3.2 THEORETICAL FRAMEWORK: MASSIVE ACTIVATIONS IMPLY COMPRESSION

We now prove that massive activations necessarily induce the observed compression. Consider the representation matrix $\mathbf{X} \in \mathbb{R}^{T \times d}$ with rows $\{\mathbf{x}_i\}_{i=0}^{T-1}$, where $\mathbf{x}_0$ denotes the BOS token.

**Theorem 1** (Massive Activations Induce Spectral Dominance). *Let $M = \|\mathbf{x}_0\|^2$, $R = \sum_{i \neq 0} \|\mathbf{x}_i\|^2$, and $\theta_i$ be the angle between $\mathbf{x}_0$ and $\mathbf{x}_i$. Define the* alignment term $\alpha = \frac{1}{R} \sum_{i \neq 0} \|\mathbf{x}_i\|^2 \cos^2 \theta_i \in [0, 1]$. *Then:*

$$\sigma_1^2 \geq M + \alpha R, \tag{3}$$

*where $\sigma_1$ is the largest singular value of $\mathbf{X}$.*

*Proof Sketch.* Full proofs and discussions in Appendix A.2. □

This theorem has immediate consequences for compression metrics:

**Corollary 2** (Compression Bounds). *Let $c = M/R$ be the norm ratio and $p = (c + \alpha)/(c + 1)$. Then:*

1. ***Dominance:*** $\sigma_1^2 / \sum_{j \geq 2} \sigma_j^2 \geq (c + \alpha)/(1 - \alpha)$

2. ***Anisotropy:*** $p_1 \geq p$

3. ***Entropy:*** $H(\mathbf{X}) \leq -p \log p - (1 - p) \log(1 - p) + (1 - p) \log(r - 1)$

---

[1] Here, Llama-3-8B behaves as an extreme outlier: almost all layers exhibit very large BoS activations, which makes the layerwise first-order differences nearly flat. This, in turn, renders the per-layer correlation ill-defined and artificially suppresses the overall correlation magnitude while inflating its variance.

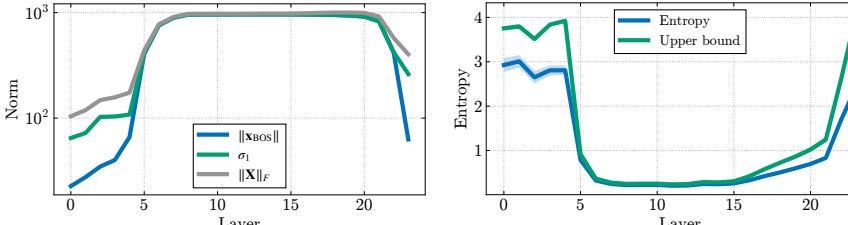

Figure 3: **Our theoretical bounds become exact when massive activations emerge, proving they drive compression. Left:** When BOS norm dominates (layers 5–15), the first singular value $\sigma_1 2$ approximately equals both $\|\mathbf{x}_{\text{BOS}}\|$ and $\|\mathbf{X}\|_F$, confirming near rank-one structure. **Right:** Our entropy upper bound (Corollary 2) tightly matches empirical values in compressed layers, validating that massive activations mathematically necessitate compression. Average across 100 RedPajama (Weber et al., 2024) examples for Pythia 410M.

**Meaning of the results.** Theorem 1 shows two factors control the rise of $\sigma_1^2$: (i) the magnitudes of the activations $M$, and (ii) the *alignment* $\alpha$ of other rows with $\mathbf{x}_0$. Full alignment makes $\mathbf{X}$ rank one (with $\sigma_1^2(\mathbf{X}) = M + R = \|\mathbf{X}\|_F^2$), while even with small $\alpha$ with large $M$ suffices to grow $\sigma_1^2$. The corollaries give lower bounds on dominance and anisotropy in terms of $(c, \alpha)$, so increasing $c$ (stronger gap between activations) or increasing $\alpha$ (stronger alignment) provably widens the spectral gap. Consequently, the singular-value entropy is tightly upper-bounded by the mass in the top component, so $c \gg 1$ or $\alpha \to 1$ also results in $H(\mathbf{X})$ dropping towards zero. Empirically, we find that $\alpha R$ remains small compared to $M$ throughout the compression regime, so the behaviour of $\sigma_1^2$ is effectively dictated by the massive activation term $M$. More details in Appendix B.1.

**Tightness of the bounds in practice.** When massive activations create growing norm ratios $c$, these bounds become tight. Figure 3 compares our theoretical bounds against empirical measurements for Pythia 410M across all layers. In early layers where massive activations are absent, the bounds are loose as expected, because the theory only constrains the intermediate layers. However, in middle layers where massive activations emerge, the bounds become nearly exact, with predicted and observed values overlapping within measurement error. This tightness reveals that massive activations are the dominant mechanism shaping representation geometry: when $\|\mathbf{x}_0\|$ becomes massive, the representation matrix effectively becomes rank-one plus small perturbations, exactly as our theory predicts.

### 3.3 EVIDENCE FROM TARGETED ABLATIONS

To isolate the exact role of massive activation empirically, we perform targeted ablations: zeroing the MLP's contribution to the BOS token at layers where massive activations emerge. Specifically, we set $\mathbf{x}_{\text{BOS}}^{(\ell+1)} \leftarrow \mathbf{x}_{\text{BOS}}^{(\ell)} + \text{Attn}^{(\ell)}(\mathbf{x}_{\text{BOS}})$, removing only $\text{MLP}^{(\ell)}(\mathbf{x}_{\text{BOS}})$.

We find that **ablating massive activations can eliminate both phenomena**, confirming our theoretical conjecture. In LLaMA3 8B, removing the MLP's contribution at the first layer prevents entropy drop (remains at 0.4-0.5 bits vs. dropping to 0.02 bits), eliminates sink formation (sink rate

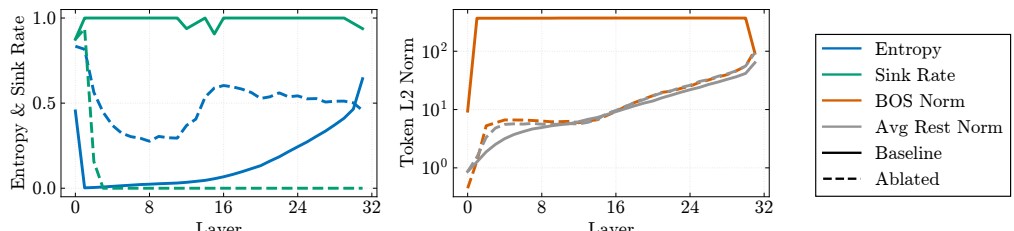

Figure 4: **Removing massive activations eliminates both compression and attention sinks, confirming causality.** Ablating the MLP contribution to the BOS token at layer 0 in LLaMA3 8B has three effects: **(Left)** Entropy remains at ∼0.5 bits instead of dropping to 0.02, showing decompression. **(Middle)** Sink rate stays at 0 throughout depth, confirming no attention sink formation. **(Right)** BOS norm (orange) remains comparable to the rest of tokens (grey) instead of spiking by $10^3\times$. This causal intervention validates that massive activations drive both phenomena.

drops from 0.85-1.0 to 0.0) and maintains BOS norm within $2\times$ of other tokens (vs. $10^2\times$ normally) (Fig. 4). Similar results hold across models, as seen in Figs.17,18 in Appendix B.1. Some models (Pythia 410M, Qwen2 7B) develop massive activations in more than one stage. Ablating any single stage partially reduces compression; ablating all stages eliminates it entirely, suggesting a cumulative contribution.

**Why Middle Layers?** We hypothesize that the mid-depth concentration of sinks and compression reflects how Transformers allocate computation across depth: early layers perform broad contextual mixing, while later layers become increasingly aligned with next-token prediction (Lad et al., 2024). Freed from these pressures, middle layers can develop extreme features that regulate token-to-token sharing and induce compression, consistent with a mid-network shift toward refinement (Csordás et al., 2025). In the next section, we show how massive activations predict, and precisely characterize, three stages of information flow.

## 4 MIX-COMPRESS-REFINE: A THEORY OF INFORMATION FLOW

> **Key Insight:** Transformers organize computation into three distinct phases demarcated by massive activations. Early layers mix information broadly to build context. Middle layers compress representations and halt mixing when massive activations emerge, preventing over-smoothing while maintaining essential information. Late layers equalize norms and switch to local positional attention, refining representations for task-specific outputs.

Building on our mechanistic understanding of massive activations, we now present a broader theory of how transformers organize computation across depth. We propose that information processing occurs in three distinct phases, demarcated by the emergence and dissipation of massive activations.

### 4.1 PHASE 1: INFORMATION MIXING (LAYERS 0–20%)

In the early layers of the model, we observe diffuse attention patterns, which are enabled by the lack of massive activations. This allows the model to perform mixing of high-dimensional token representations for a few layers, allowing the model to build contextual representations through broad information integration. An example of such an attention head in the early layers can be seen in Figure 6.

To quantify mixing in attention heads, we define the *Mixing Score* as the average row entropy of attention matrices: $\frac{1}{T}\sum_{i=1}^{T} H(\mathbf{A}_{i,:}^{(\ell,h)})$. Across models, we find that early layers consistently maintain mixing scores above $0.7$, confirming active token mixing, before dropping sharply when massive activations emerge. Notably, this mixing phase varies in extent: From just the first layer in some models to approximately 20% of network depth in others. However, its qualitative characteristics remain consistent. We plot this metric Figure 23 in Appendix B.2 across models and layers.

We believe that this initial mixing stage is deliberately limited to prevent over-mixing and representational collapse that would occur with extended uniform attention (Barbero et al., 2024; 2025a), analogous to over-smoothing in graph neural networks (Arroyo et al., 2025). This controlled, brief mixing phase establishes the semantic foundation that subsequent phases refine. The model captures both local token dependencies and global context, creating rich representations that can be selectively compressed and refined in later phases.

### 4.2 PHASE 2: COMPRESSION AND HALTED MIXING (LAYERS 20–85%)

The middle phase begins abruptly with the emergence of massive activations, typically on the BOS token. As established in Section 3.2, these massive activations necessarily induce representational compression, as well as attention sink formation (Gu et al., 2025). This phase serves as a computational shut-off switch for mixing. The attention sinks act as approximate "no-ops" (Bondarenko et al., 2023). By attending to BOS tokens with near-zero value norms, heads effectively skip their contribution while preserving the residual stream.

In these middle layers, the model refined information through the compressed residual stream, where a few dominant directions preserve high-level context while discarding redundancies. This aligns with the depth-efficiency perspective of Csordás et al. (2025), who show that mid-layers contribute less to shaping future tokens and more to stabilizing current representations. In Section 5, we show

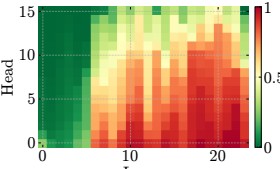 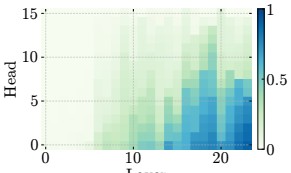 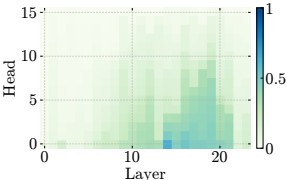

Figure 5: **Middle-layer sinks adapt to input complexity while early mixing remains constant.** **(Left)** BOS sink scores for a high-sink prompt in Pythia 410M, showing strong attention to BOS in layers 5–20. **(Middle)** Difference in BOS sink scores between high-sink and low-sink prompts, revealing input-dependent variation concentrated in middle layers. **(Right)** Difference in mixing scores between the same prompts, showing near-zero variation in early layers. This demonstrates that Phase 1 performs fixed mixing regardless of input, while Phase 2 compression dynamically adjusts sink strength based on prompt complexity.

that performance on generation tasks tends to improve mostly in the latter half of this second phase. We hypothesize this lag reflects the time mid-layer MLPs need to process and consolidate the compressed signal before yielding token-level refinements. We do not treat this as a separate phase, however, since it is not cleanly demarcated by the emergence or dissipation of massive activations.

**Sink behavior in middle to late layers adapts to input complexity.** Mid to late layer sinks are *input-dependent* ("dormant heads"), which are often inert but active on specific prompts (Guo et al., 2024). Reproducing Sandoval-Segura et al. (2025) on 20K FineWeb-Edu prompts (Lozhkov et al., 2023), we compare the "top" prompts (with the strongest sink scores) and the "bottom" prompts (with the weakest). As shown in Figure 5, sink strength diverges in the middle and last layers. This provides evidence that while middle layers default to sink-like behavior that limits mixing, sink strength in this phase varies depending on the prompt.

### 4.3 PHASE 3: SELECTIVE REFINEMENT (LAYERS 85–100%)

In the final phase, the model reverses the compression bottleneck through norm equalization and a fundamental shift in attention patterns.

**Norm equalization drives decompression.** In this phase, we find the BOS norm plateaus or decreases while the average norm of the rest of tokens rises sharply, driving them toward similar magnitudes (right panel in Figure 4 and Figure 7). This equalization begins earlier than the full phase transition, where the average norm starts rising around 40–60% depth, preparing for the eventual shift. The massive activation ratio drops from $> 10^3$ to $< 10$, removing the mathematical basis for compression and allowing representations to re-expand.

**Attention shifts to positional patterns.** As massive activations dissipate, we observe heads transition from sink-dominated to position-based patterns. In particular, we observe the emergence of *identity heads* $(i \rightarrow i)$, *previous-token heads* $(i \rightarrow i-1)$, and other sharp attention patterns, where by sharp we mean highly localized attention patterns. Figure 6 shows an example of such a pattern in the

| Layer 0, Head 3 | Layer 16, Head 4 | Layer 23, Head 10 |

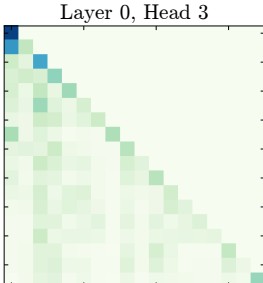 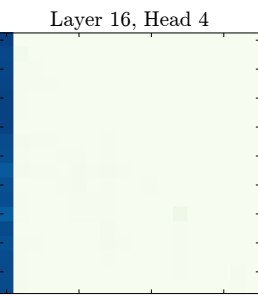 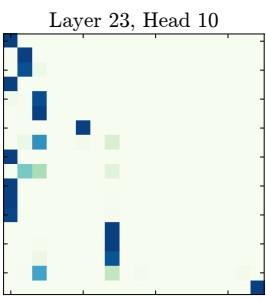

Figure 6: **Attention patterns transform from diffuse mixing to sinks to positional focus across depth.** Evolution of attention patterns in Pythia 410M showing representative heads at layers 0, 16, and 23. Early layers exhibit diffuse attention enabling broad information mixing. Middle layers show sink patterns that halt mixing. Late layers display sharp positional patterns for selective refinement.

Pythia 410M model. We find that sharp positional patterns are especially common in RoPE-based models, consistent with recent work (Barbero et al., 2025b) showing that RoPE induces frequency-selective structure that favors the emergence of such heads. We provide empirical evidence of this in Appendix B.2. In particular, when measuring the mixing rate of attention patterns, only models *without* RoPE revert to higher mixing in later layers, whereas RoPE-based models consistently transition toward sharp positional attention.

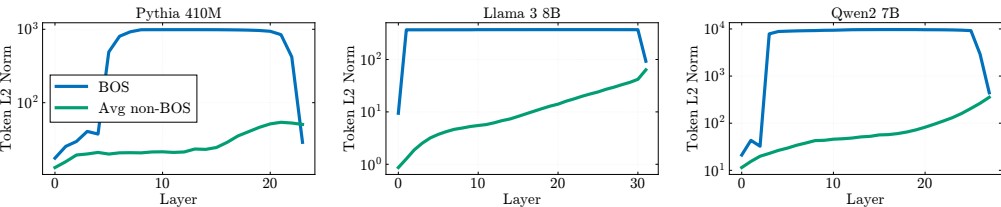

Figure 7: LLMs equalize token norms towards at the end of the network.

**This phase serves three computational purposes.** We believe that this phase serves three purposes. First, *norm equalization* reduces BOS dominance so content tokens can meaningfully influence the residual pathway and receive token-specific refinements. Second, attention shifts to *sharp* (often positional) heads that perform *selective mixing*, focusing on a few task-relevant tokens and writing their features into the residual. As late layers re-expand capacity, these signals can be represented distinctly rather than squeezed by the mid-layer bottleneck. In parallel, *identity/near-diagonal* heads curb mixing without defaulting to sinks, where their non-zero value writes act as local signal boosters, in contrast to BOS sinks that effectively zero out updates. Third, bringing token norms to smaller, comparable scales likely improves numerical stability for the unembedding. Notably, models tend to equalize by *boosting content-token norms* rather than fully collapsing the BOS norm, preserving a modest global bias while enabling precise, content-driven refinements.

## 5 IMPLICATIONS FOR DOWNSTREAM PERFORMANCE

> **Key Insight:** Different tasks achieve peak performance at different phases of the Mix-Compress-Refine organization. Embedding tasks peak during Phase 2's compression, benefiting from lower-dimensional spaces. Generation tasks improve monotonically through all phases, requiring Phase 3's refinement for accurate next token prediction. Multiple-choice reasoning shows flat performance until mid-depth, suggesting it needs both compression and subsequent refinement. This explains why studies reach different conclusions about "optimal" layers: they're measuring fundamentally different computational objectives.

Prior work (Skean et al., 2025) found that mid-layer representations perform strongly, particularly on embedding benchmarks, and linked this effect to the mid-depth compression valley. In this section, we broaden the picture by evaluating both embedding and generation tasks, relating their depthwise performance to the three-stage framework introduced above.

### 5.1 THE DISTINCT PERFORMANCE PATTERNS ACROSS TASKS

**Generation improves monotonically through all phases.** We begin by evaluating intermediate layers across multiple model families and sizes using LogitLens (Nostalgebraist, 2020) on `WikiText-2` (Merity et al., 2016). We observe a steady perplexity decline with depth from $> 10^4$ in early layers to 10-25 at full depth, as shown in Figure 8 (left). We notice little gain in the very early layers (consistent with an embedding-formation stage) and continued refinement through mid-depth. Across several models, the sharpest improvements occur in Phase 3, where norm equalization and positional/identity heads enable token-specific refinements, which appear to be crucial for next-token prediction tasks.

We further test the same set of models on multiple-choice question-answering tasks. We evaluate on ARC Easy, ARC Challenge, HellaSwag, and WinoGrande (Clark et al., 2018; Zellers et al., 2019; Sakaguchi et al., 2021) via LogitLens and the LM Evaluation Harness (Gao et al., 2024) with zero-shot learning. Figure 8 (middle) shows the results for ARC Easy and results for the rest of the

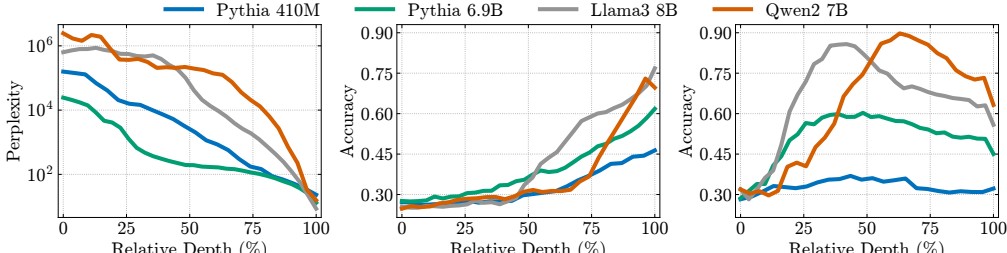

Figure 8: **Embedding tasks peak during compression while generation requires full refinement, revealing distinct computational objectives. (Left, Middle)** Perplexity on Wikitext-2 and multiple-choice QA accuracy in ARC Easy via LogitLens generally do not improve significantly until ∼50% depth, then decreases/rises steadily through Phase 3. **(Right)** Linear probe test accuracy on the same task peaks at 25–75% depth (Phase 2) and declines thereafter. This divergence demonstrates that embedding-relevant features concentrate in compressed middle layers, while generation tasks require full-depth for token-specific predictions.

datasets can be found in Figure 29, Appendix B.3. For sufficiently large models, accuracies remain largely flat until roughly 40-60% depth and then rise sharply. This suggests that, for generation-aligned tasks, *compression alone* (Phase 2) is not sufficient; gains emerge only toward the end of Phase 2 (after sufficient residual refinement) and continue into Phase 3, where norm equalization and positional/identity heads enable token-specific updates.

In short, in generation settings, the late Phase 2 to Phase 3 transition is pivotal, aligning with the observation in Csordás et al. (2025) of a mid-network phase change from future-token to current-token computation, providing independent validation that Phase 2 and Phase 3 serve distinct computational roles.

**Embedding tasks peak in middle layers.** To highlight the difference between generation and embedding tasks, we implement the following linear probing experiment. For each dataset, we encode the examples with the frozen backbone and extract hidden states at every layer. We then train a linear classifier at each layer on the training split and evaluate it on the test split. We probe the same multiple-choice QA benchmarks as before and, additionally, a sentence classification dataset (SST-2; Socher et al. 2013), assessing how much task-relevant information is linearly accessible at different depths. Figure 8 (right) shows the results for ARC Easy, highlighting the difference in optimal depths with the corresponding generation task, while full results for this experiment are shown in Fig. 31, Appendix B.3. Moreover, we reproduce Skean et al. (2025) on 32 MTEB tasks (Muennighoff et al., 2022) across a broader set of larger, decoder-only models. Across the board, we find that performance peaks consistently at 25–75% relative depth, outperforming early/late layers by 10-20% and precisely aligning with Phase 2, where compression is strongest (see Fig. 32 in Appendix B.3). These results align with evidence that next-token pretraining does not uniformly benefit perception-style classification (Balestriero & Huang, 2024). Together, they underscore *task dependence*: classification-relevant linear features concentrate in intermediate layers, whereas late layers are repurposed for token-specific generative refinement. Furthermore, the pattern suggests that massive activations in the residual pathway not only curb over-mixing via sink formation but also act as a mechanism the model uses to compress information.

## 5.2 WHY DO DIFFERENT TASKS NEED DIFFERENT PHASES?

We believe these findings clarify which tasks actually benefit from compression. In particular, embedding-style objectives (such as clustering, retrieval, classification, bitext mining, etc.) gain from Phase 2's compression because they target low-dimensional structure while discarding irrelevant information, echoing classic arguments on the benefits of information bottlenecks and compressed representations (Shwartz-Ziv et al., 2018; Kawaguchi et al., 2023). This picture aligns with evidence that LLMs produce surprisingly linear (and often linearly separable) embeddings (Razzhigaev et al., 2024; Marks & Tegmark, 2023). In particular, when features concentrate in a low-dimensional subspace, linear probing, semantic retrieval, and related embedding tasks become easier. Moreover, such linear structure has been linked to the emergence of high-level semantic

concepts (Park et al., 2023), reinforcing our hypothesis on why mid-layer compressed states tend to work well for non-generative evaluations.

By contrast, generation and reasoning require capacity that compressed states alone cannot provide. Performance improves the most once Phase 3 norm equalization restores higher entropy, and positional heads/MLPs can refine token-specific details, which is also when we observe the models being most confident about their predictions (see Fig. 28 in Appendix B.3). In this way, the model makes use of the compressed and refined representation from Phase 2, which has captured high-level ideas and semantic concepts, and expands this into higher-dimensional space to perform token-level refinements in Phase 3.

This reconciles two views: compressed mid-layers suit embedding benchmarks, whereas next-token-prediction–aligned tasks benefit from full-depth processing. Practically, "optimal layer" selection should match phase to objective, suggesting phase-aware early exiting (Schuster et al., 2022) as a potentially promising design choice.

## 6 CONCLUSION

In this work, we revisited two puzzling phenomena in decoder-only Transformers, attention sinks and compression valleys. We began with the observation that attention sinks, compression valleys, and massive activations all emerge at the same time in language models. We then proved that a single high-norm token necessarily induces a dominant singular value, yielding low matrix-entropy and high anisotropy, and we bounded these effects quantitatively.

Building on this, we proposed a Mix–Compress–Refine theory of depth-wise computation in LLMs. In particular, we show that early layers mix broadly through diffuse attention, middle layers compress and curb mixing via attention sinks, late layers re-equalize norms and apply sharp positional heads for selective refinement. The boundaries between these phases are marked by the appearance and later disappearance of massive activations in depth. We use this organization to clarify downstream task behavior. While embedding-style tasks peak in compressed mid-layers, generation improves through late refinement and benefits from full depth.

We see this framework as a step toward a more mechanistic account of how LLMs allocate computation across depth. We hope these insights help connect head-level mechanisms with representation geometry, ultimately guiding more efficient and controllable LLM designs.

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

# A PROOFS

## A.1 ARCHITECTURE

In this work, we study decoder-only Transformers (Radford et al., 2018), which employ causal masking in attention and constitute the dominant architecture in today's large language models (Gemma Team et al., 2024; Dubey et al., 2024). We follow the notation of Barbero et al. (2024), but we importantly also consider a model with $H \geq 1$ attention heads:

$$\mathbf{z}_i^{(\ell,h)} = \sum_{j \leq i} \alpha_{ij}^{(\ell,h)} \mathbf{W}^{(\ell,h)} \mathbf{x}_j^{(\ell)}, \text{ with } \alpha_{ij}^{(\ell,h)} = \frac{\exp\left(k\left(\mathbf{q}_i^{(\ell,h)}, \mathbf{k}_j^{(\ell,h)}, \mathbf{p}_{ij}\right)\right)}{\sum_{w \leq i} \exp\left(k\left(\mathbf{q}_i^{(\ell,h)}, \mathbf{k}_w^{(\ell,h)}, \mathbf{p}_{iw}\right)\right)}$$

$$\mathbf{z}_i^{(\ell)} = \mathbf{W}^{(\ell)} \bigoplus_{h \in H} \mathbf{z}_i^{(\ell,h)} + \mathbf{x}_i^{(\ell)},$$

$$\mathbf{x}_i^{(\ell+1)} = \boldsymbol{\psi}^{(\ell)}\left(\mathbf{z}_i^{(\ell)}\right) + \mathbf{z}_i^{(\ell)},$$

where we will also denote by the matrices $\mathbf{A}^{(\ell,h)}$ the attention heads given by $\mathbf{A}_{ij}^{(\ell,h)} = \alpha_{ij}^{(\ell,h)}$. The causal masking translates into $\mathbf{A}^{(\ell,h)}$ being lower-triangular and the row-wise softmax implies row-stochasticity.

## A.2 THEORETICAL RESULTS

This section includes the proofs of the statements of Section 3.2, where we show massive activations imply the dominance of a singular value. One can obtain a weaker version of the bound focused only on the massive activation (no alignment terms) that entails weaker bounds for the spectral metrics. The following lemma serves as a proof for the fact that $\sigma_1^2(\mathbf{X}) = \max_{||\mathbf{v}||=1} ||\mathbf{X}\mathbf{v}||^2$.

**Lemma 3.** *Let* $\mathbf{A}$ *be a real symmetric* $n \times n$ *matrix with (real) eigenvalues* $\lambda_{\max} = \lambda_1 \geq \lambda_2 \geq \ldots \geq \lambda_n = \lambda_{\min}$ *and let*

$$R(\mathbf{A}, \mathbf{x}) = \frac{\mathbf{x}^\top \mathbf{A} \mathbf{x}}{\mathbf{x}^\top \mathbf{x}}$$

*denote the Rayleigh Quotient for* $\mathbf{A}$ *and a real non-zero vector* $\mathbf{x} \in \mathbb{R}^n$. *Then* $R(\mathbf{A}, \mathbf{x}) \in [\lambda_{\min}, \lambda_{\max}]$, *achieving each bound at the corresponding eigenvectors* $\mathbf{v}_{\min}, \mathbf{v}_{\max}$.

*Proof.* Let $\mathbf{A} = \mathbf{Q}^\top \boldsymbol{\Lambda} \mathbf{Q}$ be the diagonalization of $\mathbf{A}$ in the eigenbasis given by the $\mathbf{v}_i$ and let $\mathbf{y} = \mathbf{Q}\mathbf{x}$, such that $\mathbf{x} = \mathbf{Q}^\top \mathbf{y}$ for $\mathbf{Q}$ is orthogonal (i.e. $\mathbf{Q}^\top = \mathbf{Q}^{-1}$). Then,

$$R(\mathbf{A}, \mathbf{x}) = \frac{(\mathbf{x}^\top \mathbf{Q}^\top) \boldsymbol{\Lambda} (\mathbf{Q}\mathbf{x})}{\mathbf{x}^\top \mathbf{x}} = \frac{\mathbf{y}^\top \boldsymbol{\Lambda} \mathbf{y}}{\mathbf{y}^\top (\mathbf{Q}\mathbf{Q}^\top) \mathbf{y}} = \frac{\mathbf{y}^\top \boldsymbol{\Lambda} \mathbf{y}}{\mathbf{y}^\top \mathbf{y}} = \frac{\sum_{i=1}^n \lambda_i y_i^2}{\sum_{i=1}^n y_i^2} = \sum_{i=1}^n \lambda_i \left(\frac{y_i^2}{\sum_j y_j^2}\right).$$

Since the weights $w_i = y_i^2 / \sum y_j^2$ satisfy $w_i \geq 0$ and $\sum_i w_i = 1$, $R(\mathbf{A}, \mathbf{x})$ is a convex linear combination of the eigenvalues and therefore $\lambda_{\min} \leq R(\mathbf{A}, \mathbf{x}) \leq \lambda_{\max}$, with equalities when $\mathbf{x} = \mathbf{v}_{\max}, \mathbf{v}_{\min}$. $\square$

We now prove that the emergence of massive activations in some layers directly implies that the first singular value dominates the distribution, which translates into extreme values for anisotropy and matrix-based entropy. The intuition is that entropy and anisotropy are *representation-only* properties: they depend solely on the singular-value spectrum of the representation matrix $\mathbf{X}$, whose rows are token-wise representations $\{\mathbf{x}_i\}$ and columns are features. A *massive activation* means that one row, say $\mathbf{x}_0$, carries disproportionately large norm $M = \|\mathbf{x}_0\|^2$ compared to the rest of token representations, sometimes orders of magnitude larger. Let $\mathbf{v} = \mathbf{x}_0/\|\mathbf{x}_0\|$ be the direction of $\mathbf{x}_0$, notice we can always write $\mathbf{X} = \mathbf{e}_1 \mathbf{x}_0^\top + \mathbf{Y} = \mathbf{e}_1 M \mathbf{v}^\top + \mathbf{Y}$, where $\mathbf{Y}$ contains the rest of the representations. If $M$ is large compared to $\|\mathbf{Y}\|_F^2$, then $\mathbf{X}$ is effectively a rank one matrix plus a small perturbation, and we would expect $\sigma_1^2(\mathbf{X}) \approx M$ and $\mathbf{v}$ to be close to the first right singular vector. This is exactly the mechanism exploited by PCA (Maćkiewicz & Ratajczak, 1993): the first principal component points in the direction that explains the largest variance; a massive activation creates such a dominant variance direction by construction. Therefore, even before formal bounds,

we should expect $\sigma_1^2$ to dominate whenever (i) the norm ratio $c = ||\mathbf{x}_0||^2 / \sum_{i \neq 0} ||\mathbf{x}_i||^2$ is large or (ii) the remaining rows $\{\mathbf{x}_i\}_{i \neq 0}$ are measurably aligned with $\mathbf{x}_0$. The next result formalizes this intuition.

**Theorem 4.** *Let* $M = ||\mathbf{x}_0||^2$, $R = \sum_{i \neq 0} ||\mathbf{x}_i||^2$, *and* $\theta_i$ *be the angle between* $\mathbf{x}_0$ *and* $\mathbf{x}_i$. *Define the alignment term* $\alpha = \frac{1}{R} \sum_{i \neq 0} ||\mathbf{x}_i||^2 \cos^2 \theta_i \in [0, 1]$. *Then:*

$$\sigma_1^2 \;\geq\; ||\mathbf{x}_0||^2 + \sum_{i \neq 0} ||\mathbf{x}_i||^2 \cos^2 \theta_i \;=\; M + \alpha R.$$

*Proof.* By definition of the singular value (also see Lemma 3),

$$\sigma_1^2 \geq \frac{\mathbf{x}_0^\top \mathbf{X}^\top \mathbf{X} \mathbf{x}_0}{\mathbf{x}_0^\top \mathbf{x}_0} \;=\; \frac{||\mathbf{X}\mathbf{x}_0||^2}{||\mathbf{x}_0||^2} \;=\; \frac{1}{||\mathbf{x}_0||^2} \sum_{i=0} \langle \mathbf{x}_i, \mathbf{x}_0 \rangle^2 \;=\; ||\mathbf{x}_0||^2 + \sum_{i \neq 0} \frac{\langle \mathbf{x}_i, \mathbf{x}_0 \rangle^2}{||\mathbf{x}_0||^2}.$$

Using $\langle \mathbf{x}_i, \mathbf{x}_0 \rangle^2 = ||\mathbf{x}_0||^2 ||\mathbf{x}_i||^2 \cos^2 \theta_i$, we obtain

$$\sigma_1^2 \geq ||\mathbf{x}_0||^2 + \sum_{i \neq 0} ||\mathbf{x}_i||^2 \cos^2 \theta_i$$

Since $\alpha R = \sum_{i \neq 0} ||\mathbf{x}_i||^2 \cos^2 \theta_i$, we get $\sigma_1^2 \geq M + \alpha R$, which is the desired result. $\square$

As mentioned in the main text, Theorem 4 makes precise how two independent factors govern the rise of $\sigma_1^2$: (i) the magnitudes of the activations $M$, and (ii) the *alignment* $\alpha$ of the remaining rows with $\mathbf{x}_0$. If representations were totally aligned, then $\mathbf{X}$ would indeed be rank one and would have one singular value given by $\sigma_1^2(\mathbf{X}) = M + R = ||\mathbf{X}||_F^2$. Conversely, even with small $\alpha$ (say, when token representations are not aligned or even orthogonal), a large norm M suffices to grow $\sigma_1^2$. Empirically, we observe the term $||\mathbf{x}_0||^2$ making the most impact in our analysis, as we know it will be orders of magnitude larger than the rest of norms, however keeping the alignment term is also important for the following results.

We move onto proving Corollary 2, which we split in three in this section.

**Corollary 5** (Singular value dominance). *In the setting of Theorem 4, let* $c = ||\mathbf{x}_0||^2 / \sum_{i \neq 0} ||\mathbf{x}_i||^2$, *then*

$$\sigma_1^2 \;\geq\; \left( \frac{c + \alpha}{1 - \alpha} \right) \sum_{j \geq 2} \sigma_j^2.$$

*Proof.* From Theorem 4, $\sigma_1^2 \geq M + \alpha R$. Moreover, $\sum_{j \geq 2} \sigma_j^2 = ||\mathbf{X}||_F^2 - \sigma_1^2 \leq ||\mathbf{X}||_F^2 - (M + \alpha R) = R - \alpha R = (1 - \alpha) R$. Therefore one gets:

$$\frac{\sigma_1^2}{\sum_{j \geq 2} \sigma_j^2} \geq \frac{M + \alpha R}{||\mathbf{X}||_F^2 - (M + \alpha R)} = \frac{M + \alpha R}{(1 - \alpha) R} = \frac{\frac{M}{R} + \alpha}{1 - \alpha} = \frac{c + \alpha}{1 - \alpha}.$$

$\square$

**Corollary 6** (Anisotropy). *Let* $p_1 = \sigma_1^2 / ||\mathbf{X}||_F^2$ *denote the anisotropy. In the setting of 4,*

$$p_1 \;\geq\; \frac{M + \alpha R}{M + R} \;=\; \frac{c + \alpha}{c + 1}.$$

*Proof.* Divide $\sigma_1^2 \geq M + \alpha R$ by $||\mathbf{X}||_F^2 = M + R$, therefore:

$$p_1 \geq \frac{M + \alpha R}{M + R} = \frac{\frac{M}{R} + \alpha}{\frac{M}{R} + 1} = \frac{c + \alpha}{c + 1}.$$

$\square$

As mentioned in the main text, Corollaries 5, 6 lower-bound the dominance ratio and anisotropy using only $(c, \alpha)$. Thus, either increasing $c$ (stronger massive activation) or increasing $\alpha$ (stronger alignment) provably inflates the spectral gap. In both cases, having perfect alignment with $\mathbf{x}_0$ or having $||\mathbf{x}_0||^2$ grow with respect to the rest, forces extreme values. If $\alpha \to 1$, then $\frac{c+\alpha}{1-\alpha} \to \infty$

and $\frac{c+\alpha}{c+1} \to 1$, intuitively because only one direction becomes relevant in the data. Moreover, as the massive activation grows $c \to \infty$, the same result holds. Notice that $c$ is the ratio between the massive activation and the rest of them, therefore $c$ increases by letting $\mathbf{x}_0$ be larger in norm, but also letting the rest of representations have low norm.

**Corollary 7** (Shannon matrix-based entropy). *Let $p_j := \sigma_j^2/\|\mathbf{X}\|_F^2$ denote the normalized distribution of singular values of $\mathbf{X}$. Let $H(\mathbf{X}) := -\sum_{j=1}^{r} p_j \log p_j$ be the Shannon entropy of such distribution. Let $p := \dfrac{c+\alpha}{c+1}$. Then, we have the following bound*

$$H(\mathbf{X}) \leq -p \log p - (1-p) \log(1-p) + (1-p) \log(r-1),$$

*Proof.* Let

$$H(\mathbf{X}) = -p_1 \log p_1 - \sum_{j=2}^{r} p_j \log p_j \leq -p \log p - \sum_{j=2}^{r} p_j \log p_j,$$

so we need to bound the second term, which is the entropy of $r-1$ terms adding up to $1-p_1 \leq 1-p$. This term would be maximised if the mass was equally distributed, that is, $p_j = \frac{1-p_1}{r-1} \leq \frac{1-p}{r-1}$. Therefore, one gets

$$-\sum_{j=2}^{r} p_j \log p_j \leq -\sum_{j=2}^{r} \frac{1-p}{r-1} \log\left(\frac{1-p}{r-1}\right) = -(1-p) \log\left(\frac{1-p}{r-1}\right).$$

The result is obtained combining these two bounds. □

For fixed top mass $p_1 \geq p$, entropy is maximized when the remaining mass $1-p$ is spread uniformly over the other $r-1$ singular values; the bound above is exactly that maximum. Consequently, any additional structure in the tail (e.g., a second spike) will lower the true entropy beneath this upper bound. Notice for $c \to \infty$ or $\alpha \to 1$, $p \to 1$ and the upper bound approaches $0$.

**Limitations of this analysis.** In the theoretical analysis conducted above, we only considered one massive activation placed on the BOS token. In practice, models may exhibit more than one massive activation (Sun et al., 2024). In this case, our $c$ term would make the bounds more permissive. We believe this poses no problem to our overall message and that this analysis can be extended. One can suppose the first $n$ tokens to be the massive activations and decompose $\mathbf{X} = \sum_{i=0}^{n-1} \mathbf{e}_i \mathbf{x}_i^\top + \mathbf{Y}$ such that the first summand can be of rank at most $n$ and $\mathbf{Y}$ a small perturbation in comparison, leading to small entropy (effective rank $\leq n$), also holding for longer context lengths.

## B    ADDITIONAL RESULTS

**Experimental details.**   All experiments were implemented in PyTorch using NVIDIA A100 GPUs with 40GB memory or NVIDIA H100 GPUs with 80GB when the memory requirements were stronger. We examined pretrained models of varying depths, using HuggingFace repositories with Transformers and Transformer-Lens (Nanda & Bloom, 2022). When large datasets were run to collect metrics such as sink rates and norms, prompts were truncated to a maximum length of 4096 tokens for the FineWeb-Edu experiment (Fig. 5) and 1024 for the GSM8K experiment (Fig. 1), as the latter required singular value decompositions to compute the entropy. LogitLens experiments for multiple-choice-questions tasks were done with LM-Evaluation-Harness (Gao et al., 2024), implementing our own model wrapper to output hidden states at each layer instead of final ones.

**Pearson Correlations.**   To assess the dynamical relationship between BOS norm, matrix-based entropy, and BOS sink rate across layers, we computed correlations on their layerwise changes. For each model and metric, the trajectory across layers was first $z$-scored, and then we defined the delta at layer $\ell$ as the difference with respect to the preceding layer,

$$\Delta\tilde{b}_\ell = \tilde{b}_\ell - \tilde{b}_{\ell-1}, \quad \Delta\tilde{e}_\ell = \tilde{e}_\ell - \tilde{e}_{\ell-1}, \quad \Delta\tilde{s}_\ell = \tilde{s}_\ell - \tilde{s}_{\ell-1}.$$

This procedure emphasizes abrupt layerwise changes rather than absolute values, which is crucial because BOS norm often exhibits sharp spikes that coincide with collapses in entropy and the subsequent emergence of attention sinks. We then measured Pearson correlation coefficients between $\Delta\tilde{b}_\ell$ and $\Delta\tilde{e}_\ell$ (BOS norm vs entropy, same layer) and between $\Delta\tilde{b}_\ell$ and $\Delta\tilde{s}_{\ell+1}$ (BOS norm vs sink rate, lagged by one layer). Correlations were computed separately per model and summarized across models by Fisher $z$-transform averaging, reporting the mean correlation and the standard deviation across models.

**Limitations.**   We outline some limitations of our work. Our analysis focuses on decoder-only Transformers and primarily attributes both sinks and compression to BOS-centered massive activations; models with alternative positional schemes, attention sparsity patterns, or special-token conventions (e.g., no explicit BOS token, sinks in different positions or ALiBi encodings) may exhibit different dynamics. Our causal claims use targeted MLP ablations on selected layers and model families, however, we observe model-dependent exceptions (e.g., sinks persisting despite decompression). Lastly, the theory assumes a single massive row, whereas real models may feature multiple interacting massive activations. However, as discussed in Appendix A.2, we believe this poses no harm to the overall message: a few massive activations would push the representations to a lower-dimensional subspace, but not necessarily of dimension 1.

### B.1    BROADER ANALYSIS OF MODELS

In this section, we provide broader validation of our three-phase theory across model families and model sizes. Moreover, we expand on the empirical measurements of metrics from our theoretical analysis, on the ablation of MLPs and provide two notes on the specifics of the GPT OSS model and Gemma 7B.

**Validation on more datasets.**   The GSM8K dataset use in Figure 1 only contains structured question-answer pairs that we concatenate to obtain each prompt. In order to test our theory against different distributions of data, we further evaluate these models on 5K examples from the following datasets: The Stack (Kocetkov et al., 2022), for code; FineWeb (Penedo et al., 2024), for prose and the MATH dataset (Hendrycks et al., 2021), for math questions. The results are shown in Figures 9,10 and 11. We find that the results to be very similar to those of GSM8K, further reinforcing the input-independence of our theory.

**Validation on more model families and large models.**   To further validate our Mix-Compress-Refine theory we observe the emergence of compression, attention sinks and massive activations in the Pythia model family (Fig. 12) and in very large models (70B-120B), specifically LLaMA3 70B, Qwen2 72B and GPT OSS 120B (Fig. 13). The prompt is a single GSM8K example. GPT OSS' particular sink patterns are explained later in this section. We believe this showcases our observed correlations are a universal phenomena in LLMs.

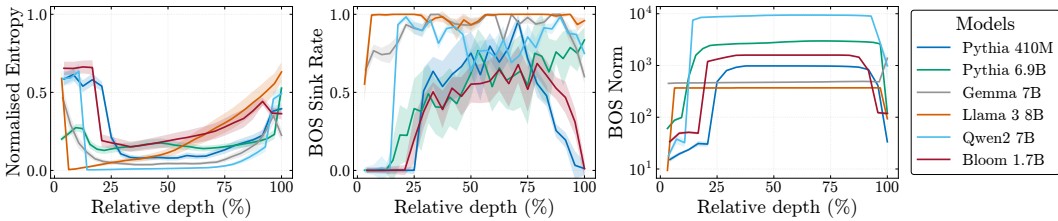

Figure 9: Entropy, sink rate and BOS in The Stack.

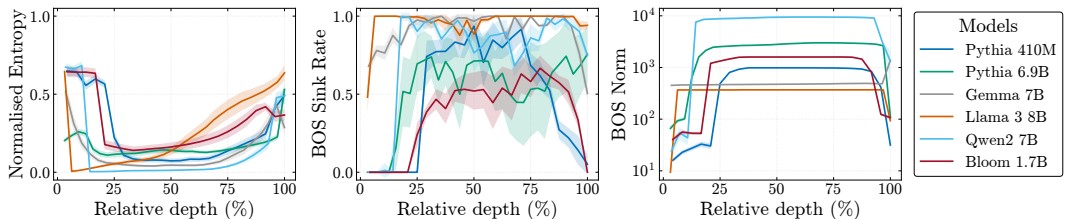

Figure 10: Entropy, sink rate and BOS in FineWeb.

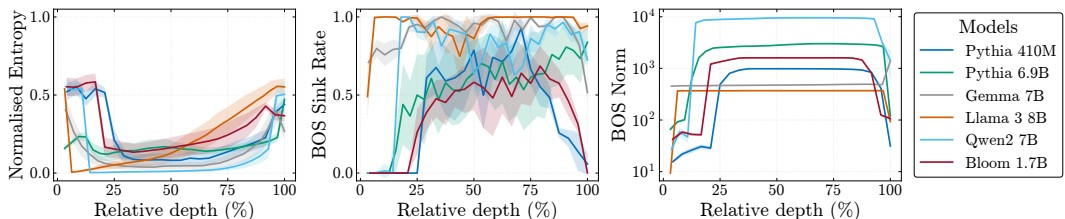

Figure 11: Entropy, sink rate and BOS in the MATH dataset.

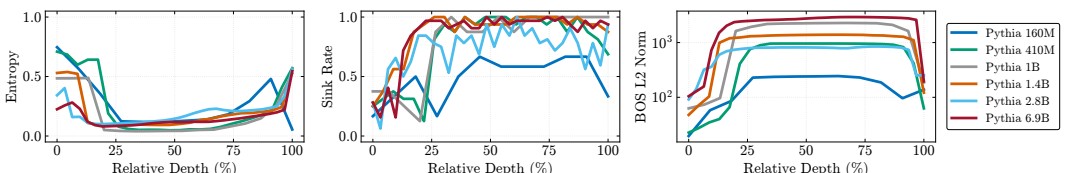

Figure 12: Entropy, sink rate and BOS norm for the Pythia family of models.

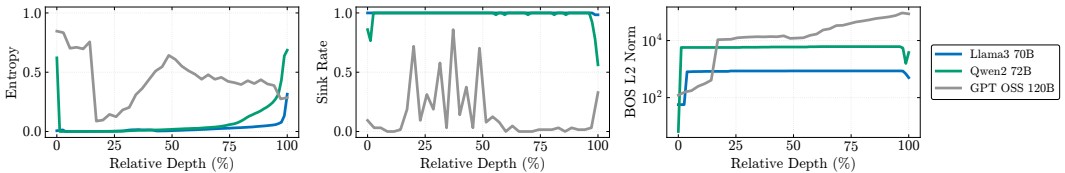

Figure 13: Entropy, sink rate and BOS norm for 70B-120B models.

**Training Checkpoints.** We evaluated the training dynamics of the Pythia 410M/6.9B/12B models across multiple checkpoints (steps 1, 1k, 2k, 4k, 8k, 10k, 20k, 30k, and 143k). At each checkpoint, after every layer we recorded the entropy, BOS sink rate (threshold $\tau = 0.3$) and the norm of the BOS token representation. The prompt was a single GSM8K prompt "`Janet's ducks lay 16 eggs...`" Figure 14 illustrates the results for the Pythia 12B model.

**Role of $\alpha$ in practice and visualization of theoretical results.** The alignment term $\alpha$ is added in the theoretical analysis for mathematical completeness. If every other token is closely aligned with the BOS token, then rank($\mathbf{X}$) $\approx 1$ and the compression follows trivially. Of course, this is not

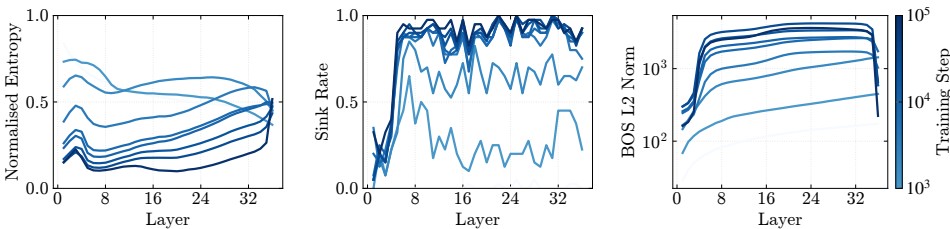

Figure 14: Entropy, Sink rate and BOS Norm at different training checkpoints for Pythia 12B.

expected to happen in practice. One can perform the same analysis without $\alpha$ using the (weaker) bound $\sigma_1^2 \geq M$ and the corresponding dominance, anisotropy and entropy bounds. The bounds in that case become $\sigma_1^2 \geq c \sum_{j=2}^{r} \sigma_j^2$, $p_1 \geq c/(c+1)$ and similarly for the entropy if one instead uses the new bound for the anisotropy. We provide plots with the bounds from the theoretical discussion in Section 3.2 and their weaker versions. Figures 15 and 16 show these values for LLaMA3 8B and Pythia 410M. We show (1) the terms $M = ||\mathbf{x}_{\text{BOS}}||^2$, $\alpha R$ and $M + R = ||\mathbf{X}||_F^2$ from Theorem 1, (2) the top 3 singular values $\sigma_i^2$ and the sum $\sum_{i \geq 1} \sigma_i^2$ and (3,4,5) the dominance, anisotropy and entropy bounds from Corollary 2. In all cases, we observe the bounds being tight in the middle layers. In this regime, the first singular value $\sigma_1^2$ follows the trajectory of $||\mathbf{x}_{\text{BOS}}||^2$ closely and dominates the rest of the singular values. The dominance decreases steadily, specially towards the second half of the network, indicating the preparation for next token prediction of Phase 3. Lastly, one observes that the weaker version of the bounds is already very tight, showing the importance of the massive activation.

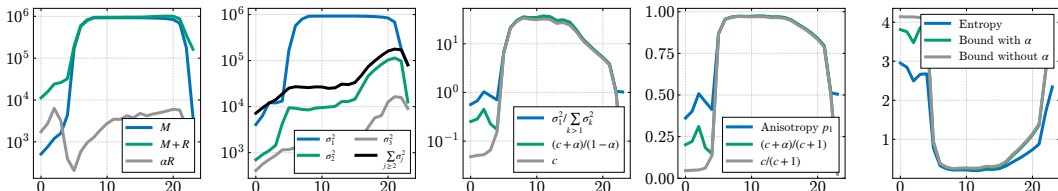

Figure 15: Theoretical bounds for Pythia 410M. **(a)** shows the elements in the bound of Theorem 1. **(b)** shows the singular value spectrum of $\mathbf{X}^{(\ell)}$ at each layer $\ell$. For plots **(c,d,e)** gray represents the weaker bound (no $\alpha$), while green represents the bound as presented in Corollary 2.

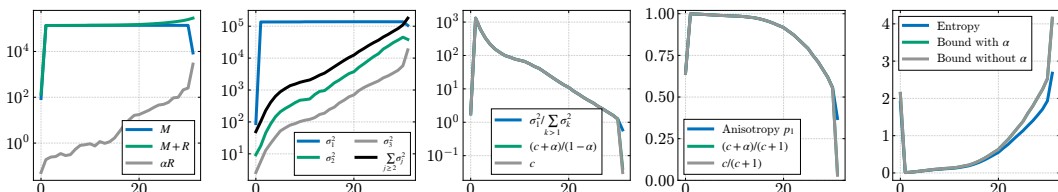

Figure 16: Theoretical bounds for LLaMA3 8B. **(a)** shows the elements in the bound of Theorem 1. **(b)** shows the singular value spectrum of $\mathbf{X}^{(\ell)}$ at each layer $\ell$. For plots **(c,d,e)** gray represents the weaker bound (no $\alpha$), while green represents the bound as presented in Corollary 2.

**MLP ablations.** We further run the targeted MLP ablations on more models to erase the appearance of the massive activation. For LLaMA3 8B, we ablate layer 0; for Qwen2 7B, we ablate layers 3 and 4 and for Pythia 410M, we ablate layers 0 and 5-7. The results are shown in Figures 4, 17, 18.

**Additional Ablations on Massive Activations** Here, we also run ablation regarding the emergence and exact effect of massive activation on both compression and attention sinks on Pythia 410M. In particular, Figure 19 shows what components of the model cause the emergence of a massive activation on the BoS token. On the other hand, Figure 20 shows the effect of clipping the maximum value in the massive activation vector to be between $[-\gamma, \gamma]$.

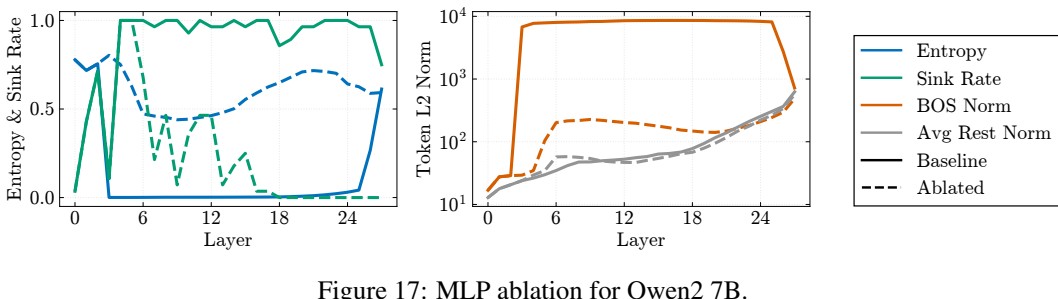

Figure 17: MLP ablation for Qwen2 7B.

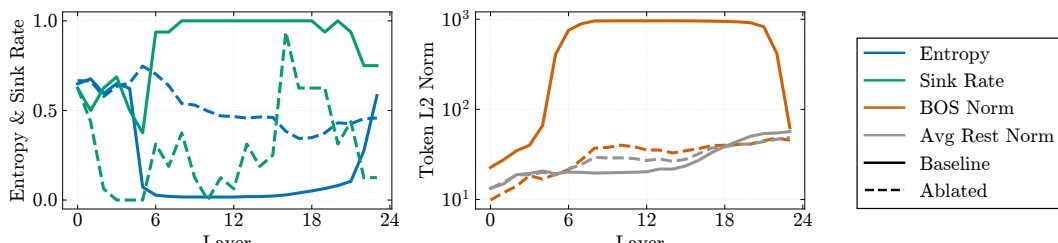

Figure 18: MLP ablations for Pythia 410M.

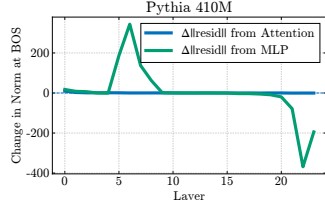

Figure 19: MLPs are responsible for the appearance of massive activations in Pythia 410M.

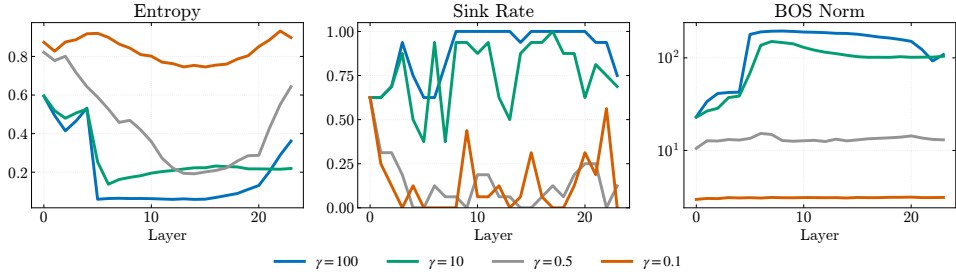

Figure 20: Ablation of the effects of massive activations in the residual pathway on Pythia 410M. Lower massive activation norms generally correspond to both increased compression and lower sinks.

From the Figures, one can see that the MLP seems to be causing the emergence of massive activations and how lowering the norm of the massive activation corresponds to an increase the in the representational entropy and decrease in attention sinks. In general, we find attention sinks to be less predictable than entropy (perhaps due to the way the sink score is calculated), although there is a general downward trend as the massive activation norm is reduced.

**A note on GPT OSS.** In the GPT OSS (Agarwal et al., 2025) family of models, each attention head is equipped with a learnable sink logit that allows it to divert probability mass away from real tokens, effectively providing a "skip" option. However, unlike the explicit $(k', v')$ bias formulation studied in Sun et al. (2024); Gu et al. (2025), GPT OSS does not include a learnable value sink token. This means the model cannot encode bias information directly through the sink, and we hypothesize that

it instead continues to rely on massive activations at the BOS token to implement bias-like behavior and generate compression. This explains why BOS sink patterns are still observed, particularly in the middle layers (see Figure 21). The alternating spikes across layers may be a consequence of GPT OSS' alternating dense and locally banded sparse attention pattern: in layers with local attention windows, heads are less able to access BOS, while in subsequent dense layers BOS becomes globally visible again, producing the observed oscillatory sinkness.

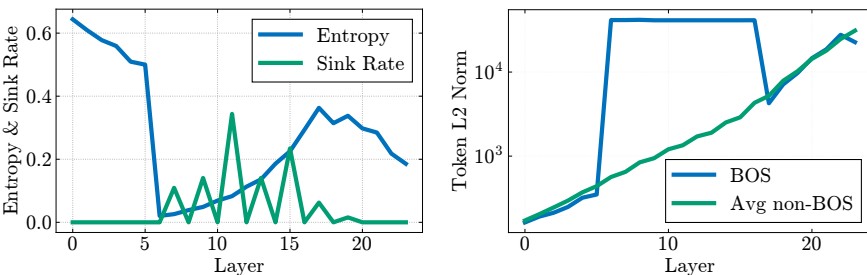

Figure 21: Entropy, sinks, and massive activations for GPT OSS 20B.

**The particular case of Gemma 7B.** Even though Gemma 7B follows the same dynamics we have discussed in the chapter, how it achieves them is different from the rest. Token norms in Gemma 7B start very high; instead of increasing the BOS norm to create a massive activation, Gemma 7B decreases the norms of the remaining tokens to create the disparity needed for compression, then re-equalizes by increasing their norms in late layers. We attribute the initially high norms to the embedding layer, as there are no other components that can account for it. We believe this is also why attention patterns in Gemma 7B look a bit different from the rest, with identity heads emerging both at the early and later layers. Figure 22 illustrates this. Pre- means before each layer, while post- means after each layer.

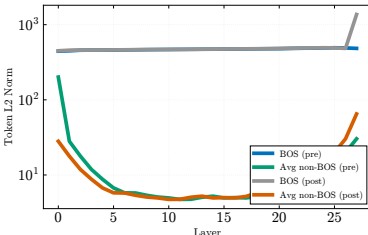

Figure 22: Token norms in Gemma 7B. The BOS norm starts high since the beginning.

## B.2 MIXING AND SINK DETECTION METRICS

In this section we propose and study new metrics for quantifying mixing and "sinkiness" in attention heads, and provide further validation on the FineWeb-Edu experiment from Section 4.2.

**Mixing Score.** Let $\mathbf{A}$ be a lower triangular, row-stochastic attention matrix. We define the *Mixing Score* as the average Shannon Entropy of each row $H_{\text{row}} = \frac{1}{T}\sum_{i=1} H(\mathbf{A}_{i,:}) = \frac{1}{T}\sum_{i=1} -\sum_{j=0}^{i} \alpha_{ij}\log\alpha_{ij}$. Since each row of $\mathbf{A}$ is the output to a $\text{softmax}$, it is a probability distribution so the score is well-defined. This captures how broadly each token is attending to its preceding tokens. High values indicate the rows are close to the uniform distribution, suggesting broad mixing across tokens. Low values imply the rows are one-hot vectors, suggesting very localized mixing (sinks, identity or positional heads). Figure 23 (right) shows the Mixing Score in depth for a variety of models, showing how the mixing abruptly decreases from 0.7-0.75 to 0.3-0.4 after the first few layers. Bloom 1.7B resumes mixing in the last phase due to not being capable of producing positional patterns, as it is the only one without rotary positional embeddings (Barbero et al., 2025b).

**ColSum Concentration.** Similar to the mixing score, column sums $c'_j = \sum_i \mathbf{A}_{ij}$ capture how much attention is received by token $j$. We get a probability distribution by normalizing to $c_j =$

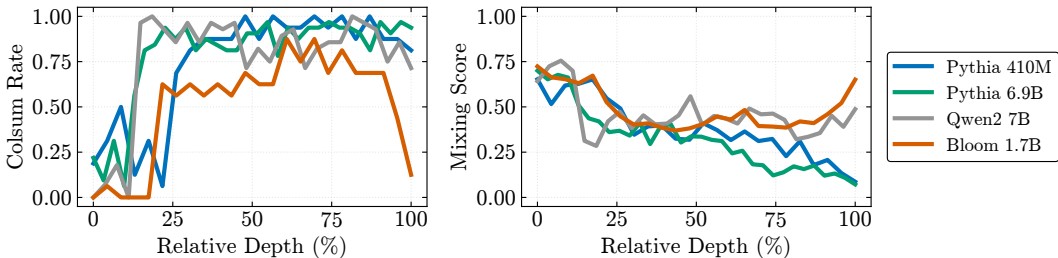

Figure 23: **Left:** ColSum Rate ($\tau = 0.3$) across depth for different models. **Right:** Mixing Score across depth for different models, averaging across heads per layer. The ColSum Rate increases with the massive activations, similar to the BOS sink rate, while the Mixing Score abruptly decreases after the first few layers.

$c_j' / \sum_i c_i' = c_j'/T$, since $\sum_i c_i' = \sum_i \sum_j \mathbf{A}_{ij} = T$ for $\mathbf{A}$ is row-stochastic. Denote by $H_{\text{col}} = -(\log T)^{-1} \sum c_j \log c_j \in [0, 1]$ the normalized entropy of such distribution. For consistency, we define the *ColSum Concentration* as $C = 1 - H_{\text{col}} \in [0, 1]$. High $C$ means a few columns receive most mass (sink-like), low $C$ means diffuse reception. When the sink is the BOS token, the ColSum Concentration is tightly related to the BOS sink score coupled: for a single BOS-dominated head, ColSum increases monotonically with the BOS score $c_0 = \frac{1}{T} \sum_i A_{i0}$ and is lower-bounded by the case where the remaining mass is spread uniformly across the other $T-1$ columns. In that case, $C_{\min}(c_0) = 1 - \left[ -(\log T)^{-1} \left( c_0 \log c_0 + (1 - c_0) \log\left(\frac{1-c_0}{T-1}\right) \right) \right]$, and any additional concentration on non-BOS columns pushes $C$ above this curve. Similar to the sink rate, we can define a *ColSum Rate* as the percentage of heads with ColSum Concentration above a certain threshold. Figure 23 (left) shows the ColSum Rate ($\tau = 0.3$) for different models across depth, imitating the Sink Rate's behavior. Moreover, scatter plots in Figure 24 show the ColSum Concentration as a function of the BOS score for all heads in different models. As given by the bound, high BOS score means high $C$, these are pure BOS sinks. Points with high ColSum but low $c_0$ reveal heads that sink to non-BOS tokens. In Pythia 410M, we observe such an outlier head, indicating a sink token different from BOS.

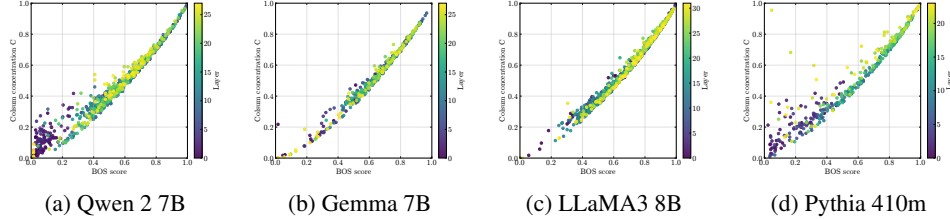

(a) Qwen 2 7B      (b) Gemma 7B      (c) LLaMA3 8B      (d) Pythia 410m

Figure 24: BOS score versus ColSum Concentration. The relationship with the BOS sink score indicates ColSum concentration is a good, token-agnostic alternative for sink detection.

**Limitations of Mixing Score and ColSum Concentration** Each of our diagnostics highlights one axis of behavior while missing others. The ColSum Concentration $C = 1 - H_{\text{col}}$ is effective at flagging sinks, where one column dominates, but it assigns zero score to identity heads and very low score to perfectly uniform heads. Conversely, the Average Row Entropy $H_{\text{row}}$ measures sparsity of rows, distinguishing diffuse mixing from one-hot attention, but it cannot differentiate which sharp pattern occurs: sinks, identities, or previous-token heads all have similarly low row entropy. Thus neither metric alone fully separates the regimes of interest. In principle, one could combine them into a scalar $\text{Mix2D}(\alpha) = \alpha C + (1 - \alpha) H_{\text{row}}$, where, for a suitable choice of $\alpha$, sinks would map near 1, perfectly uniform heads near 0, and identities near 0.5. This would give a single axis interpolating between mixing, sinkness, and identity. In practice, however, we did not find this construction very informative and thus did not include it.

**Sink-Versus-Identity Index.** In Phase 3, we observe attention patterns changing to more localized, sharp ones. Some of these patterns include identity-like heads, previous-token heads and hybrid sink-identity heads. We quantify this transition using the sink-versus-identity index, defined

as $SVI = B/(B + D)$ where $B$ is BOS attention and $D = \frac{1}{T}\sum_i \mathbf{A}_{ii}$ is diagonal attention. Therefore $B + D = \sum_{i=1} \mathbf{A}_{i0} + \mathbf{A}_{ii} \in [0, 1]$. Figure 25 plots each head as a 2D point $(SVI, B + D)$, with color corresponding to its layer. Early heads tend to have low $B + D$, indicating no attention is allocated to the BOS token nor the identity. As depth progresses, heads tend to go toward high $B + D$ and high $SVI$, indicating strong sink presence. Moreover, the middle to late layers tend to also show identity patterns or sink-identity hybrid patterns.

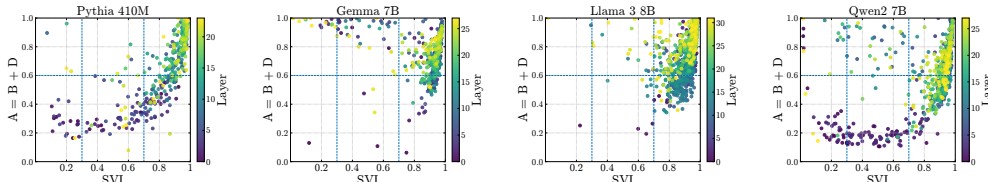

Figure 25: Attention received by the BOS token and the diagonal for each head.

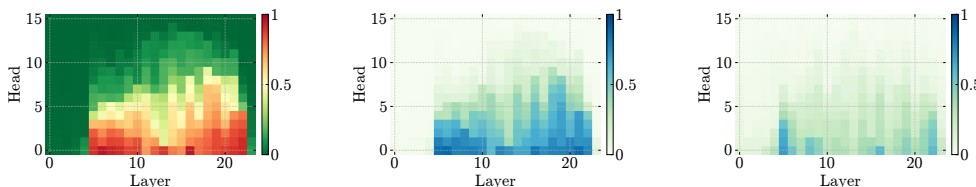

Figure 26: **Left.** BOS sink scores (top prompt, Bloom 1.7B). **Middle.** Top–bottom prompt difference in BOS sink score. **Right.** Top–bottom prompt difference in mixing score.

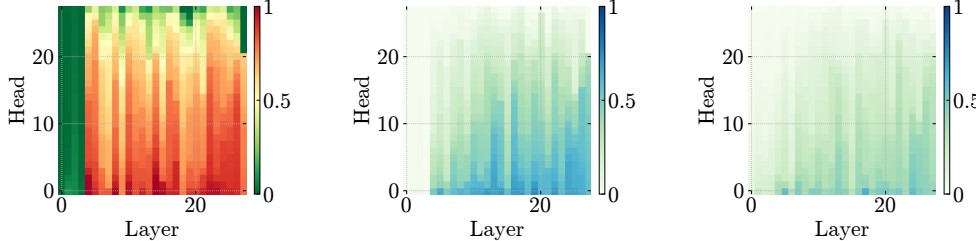

Figure 27: **Left.** BOS sink scores (top prompt, Qwen2 7B). **Middle.** Top–bottom prompt difference in BOS sink score. **Right.** Top–bottom prompt difference in mixing score.

**Further validation on FineWeb-Edu.** Figures 26 and 27 show the FineWeb-Edu experiment for Bloom 1.7B and Qwen2 7B models. The trend is clear: regardless of the input, the models do not allocate attention to the BOS token until the massive activation emerges. The amount of sinks present in the middle layers is input-dependent, however the amount of mixing performed in the early layers is not.

### B.3 ADDITIONAL RESULTS ON DOWNSTREAM TASKS

In this section we provide more details on the experiments and results exposed in Section 5 of the main text.

**Generation Tasks.** We applied the LogitLens to WikiText-2 by passing each batch of tokenized blocks through a frozen backbone and, for every layer, projecting that layer's hidden states to vocabulary logits using the model's tied unembedding head. For each layer $\ell$, we computed the next-token cross entropy loss and perplexity (as shown in Figure 8 of the main text), as well as the mean token entropy of the softmaxed logits (Ali et al., 2025), as shown in Figure 28. We take this entropy as a proxy of the model's confidence over the next token, and we also observe it decreases more rapidly towards Phase 3. In addition to next-token prediction, we extended the LogitLens evaluation to multiple-choice QA benchmarks (ARC Easy, ARC Challenge, HellaSwag, WinoGrande), where the model must select among a small set of candidate answers. For each layer, we applied the final layer

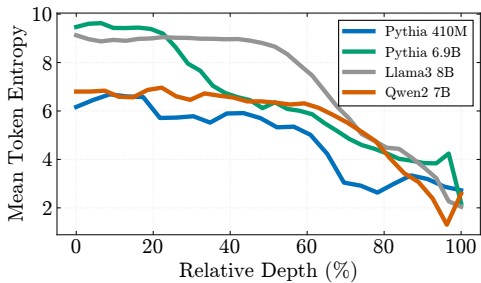

Figure 28: **Language modeling requires full depth.** Entropy of the output distribution at each layer.

norm and projected the embeddings with the tied unembedding head. We used LM Evaluation Harness to score, recording the accuracies. This allows us to compare how representations at different depths support generation-style (next-token) and selection-style (multiple-choice) reasoning. Figure 29 shows MCQ performance remains relatively flat through the compression valley of Phase 2 and begins improving towards $\sim 50\%$ of the network, underscoring that reasoning tasks require both compression and late-layer specialization. For completeness, we also ran the experiments with five-shot learning for each dataset. However, this only seemed to boost the final accuracies but did not influence the overall behavior observed.

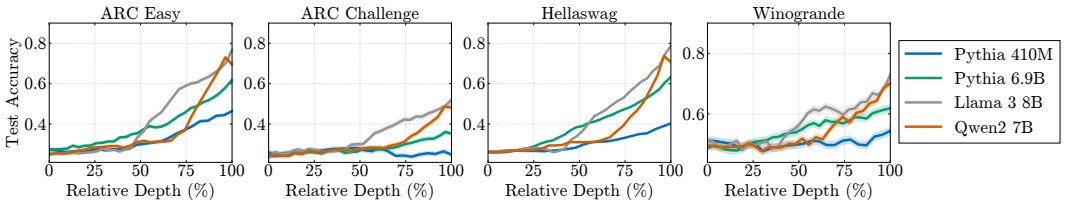

Figure 29: LogitLens accuracies on multiple choice question datasets.

**TunedLens.** TunedLens (Belrose et al., 2023) is a refinement of the LogitLens technique that involves training a small affine transformation onto the vocabulary for each layer instead of using the model's own unembedding layer. To further validate our LogitLens experiment in the MCQ datasets, we also used the LM Evaluation Harness to run the TunedLens for Pythia 410M, Pythia 6.9B and LLaMA3 8B with the pretrained lenses available at Belrose et al. (2023). We include the results in Figure 30 for completeness, however we do not observe meaningful differences in the layerwise behavior with respect to the LogitLens.

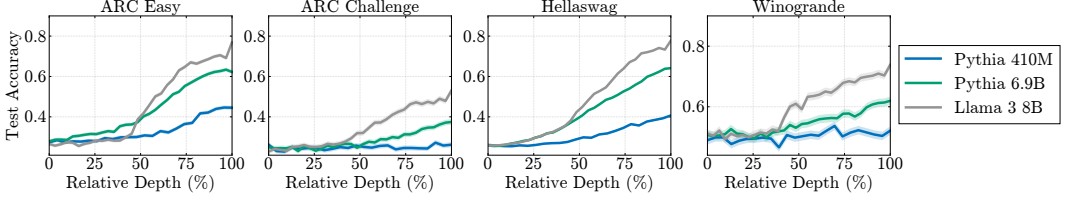

Figure 30: TunedLens accuracies on multiple choice question datasets.

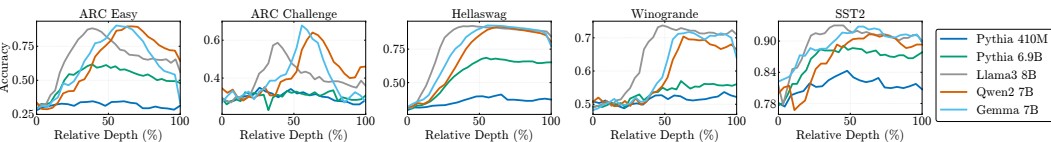

Figure 31: Linear probing validation accuracies.

**Embedding Tasks.** To further validate the results of Section 5 and the ones proposed by (Skean et al., 2025), we run a standard linear probing experiment. Probes are trained independently per layer, with backbone parameters fixed, using a learning rate of $5 \times 10^{-4}$ with PyTorch's default settings for Adam (Kingma & Ba, 2015), no weight decay, one epoch, maximum length of 1024, batch sizes of 16-32 and a random fixed seed of 123. We train for backbones Pythia 410M, Pythia 6.9B, LLaMA3 8B, Qwen2 7B and Gemma 7B. Figure 31 shows the results. As discussed, across models and datasets, accuracy peaks in the middle layers. These results suggest that the linear features relevant for classification emerge transiently in the compressed middle representations, while the late layers are repurposed for generative refinement. Moreover, we run 32 MTEB tasks for the same models and report the average main score across tasks in Figure 32.

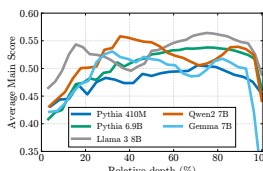

Figure 32: Average main score across 32 MTEB tasks.

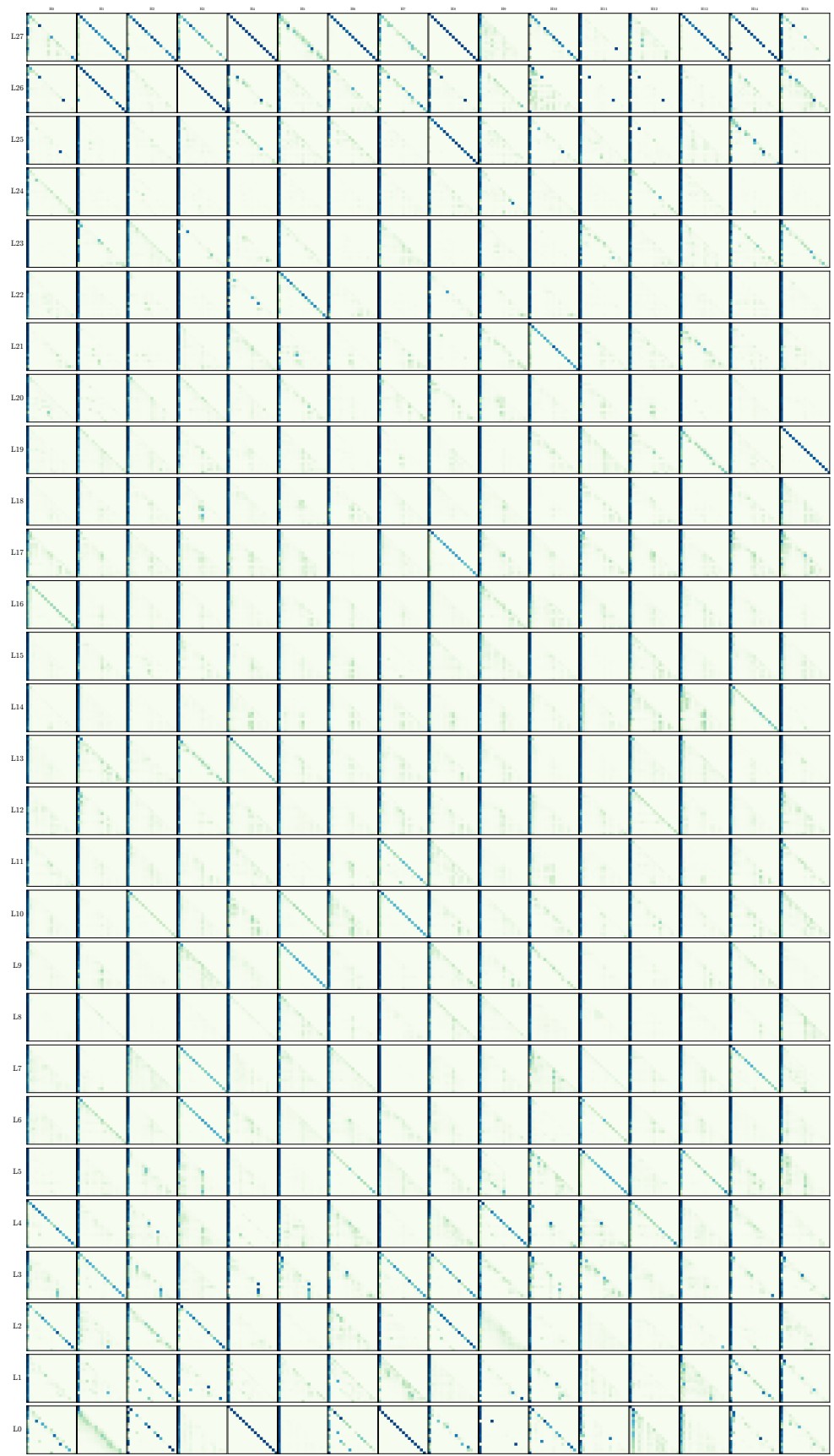

Figure 33: Gemma 7B Heads for example prompt.

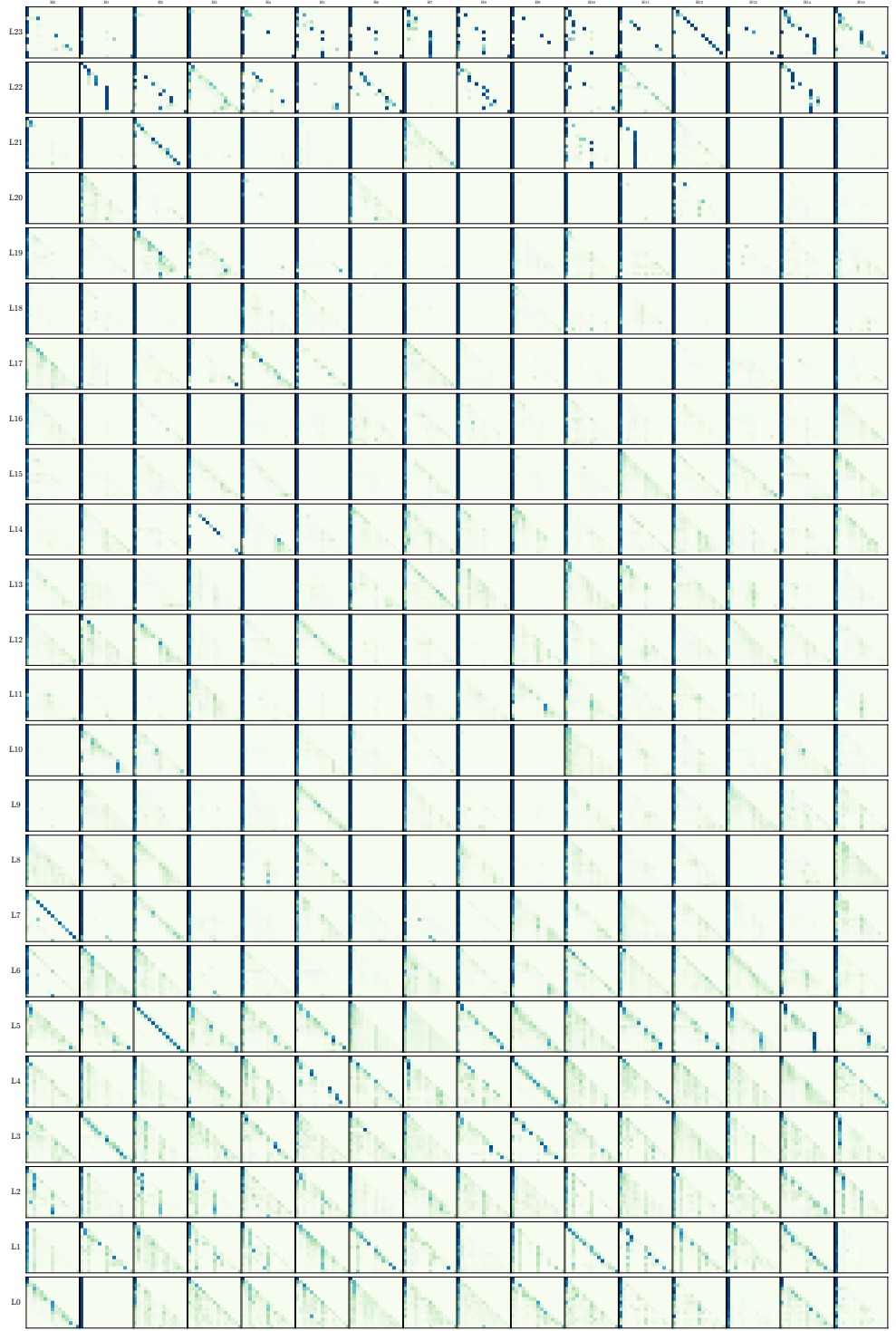

Figure 34: Pythia 410m Heads for example prompt.

