# OpenReview forum: "Attention Sinks and Compression Valleys in LLMs are Two Sides of the Same Coin"
_ICLR.cc/2026/Conference — ICLR 2026 Poster_

### Official Review · Reviewer_qBpd · 2025-10-25

**Soundness:** 3
**Presentation:** 4
**Contribution:** 4
**Rating:** 6
**Confidence:** 4

**Summary:**

The paper attempts to understand the mechanisms tying together some phenomena happening within the internal representations and activations of LLMs and interpret them in terms of information flow. Namely, they tie together massive activations (some activations in the residual stream grow extremely large), attention sinks (some attention heads put all attention on the first begin-of-string (BOS) token), and compression valleys (the representation matrix of an input at middle layers of an LLM has highly concentrated singular values, i.e., is approximately very low rank). They prove some linear algebraic bounds to show that massive activations imply compression valleys. Then, they propose a high-level framework for information flow in LLM layers called "Mix-Compress-Refine" where the initial layers mix information throughout the sequence to build context, the intermediate layers compress to preserve only essential information through the massive activations, and the final layers refine the representation for task-specific outputs. They present evidence for this through controlled experiments with real language models from 400M to 120B. They finish the paper by presenting implications for LLM-related tasks such as collecting embeddings (where one wants to collect them from the compressed but unspecialized representations in the middle layers).

**Strengths:**

- While the discussed phenomena --- attention sinks, compression valleys, massive activations --- have all been extensively explored, they have not been unified under a single explanation before. This would be a nice contribution to research trying to understand empirical phenomena in transformers.
- The theory is clear and illustrative, showing a worst-case relationship between massive activations and singular value compression. Targeted experiments show that this happens in practice, along with some training dynamics.
- The Mix-Compress-Refine framework is elegant and, if proven, would be a nice starting point for further interpretability or theoretical studies, and may be used to motivate further empirical methodology.
- The paper is nicely organized, easy to read, and has nice visualizations.

**Weaknesses:**

- The paper is not fully rigorous w.r.t. demonstrating causal mechanisms. For example, ablating MLP layers and showing that this removes massive activations and compression valleys does not show that massive activations are causally responsible for compression valleys. (Indeed, ablating MLP layers is not necessarily equivalent to ablating massive activations, as there could be several ways to do the latter.) Instead, this experiment shows that MLPs are (at least in part) a mechanism which causes massive activations and compressive valleys. Of course the theory says that both have to co-occur but it's not clear which causes the other. See "Questions" Q1 below.
- The paper discusses a lot about information flow and the information contained in some tokens, and indeed one can see this informally via the paper's examples by pointing at small vs. large weights/values in attention, etc. However, to add more rigor it may be necessary to do other tasks to confirm the precise amount of information stored in a certain location, using e.g. linear probing. See "Questions" Q2 below.
- The paper uses a single dataset GSM8K for almost all data, introducing FineWeb for a single experiment. The latter is a more diverse dataset, while the former has heavily templated questions and answers. In order to make claims at the level of generality in the paper, probably some assessments and analysis should be done on general data, and any differences/similarities with the current evaluation can be discussed and examined. See "Questions" Q3 below.

**Questions:**

Q1: Are there other mechanisms to ablate massive activations? Do they still also cause compression valleys? Can you do any kind of formal causal analysis to show the claimed causality?

Q2: Can you trace a simple end-to-end example showing the three stages, using linear probing to validate your theory of information flow and the relevant proposed mechanisms (mix, compress, refine)?

Q3: Can you compare results on different datasets? Are the results substantially different for e.g., prose vs. math questions vs code vs scientific papers (each of these is readily available from public datasets)?

---

> ### Author Response · Authors · 2025-11-21
> **Reply to the Review**
>
> We thank the reviewer for the detailed and thoughtful comments and for carefully engaging with both the theoretical and empirical parts of the paper. We found the feedback on the role of the interpretation of our causal claims particularly helpful, and these points have directly improved the revision.
>
> >  Are there other mechanisms to ablate massive activations? Do they still also cause compression valleys? Can you do any kind of formal causal analysis to show the claimed causality?
>
> Thank you for this question. Given your suggestion, and also in line with comments from Reviewers FFpb and qBpd, we performed a clipping intervention on the maximum value that can be taken in the massive activation channel. For each layer $\ell$ and clipping threshold $\gamma$, we define
>
> $Y_\ell(\gamma) := Y_\ell \text{ under } \operatorname{do}\big(\text{massive activation}^{(\ell)} \leftarrow \operatorname{clip}_\gamma(\text{massive activation}^{(\ell)})\big)$,
>
> where the clipping clamps the massive activation to the interval $[-\gamma, \gamma]$, and $Y_\ell$ denotes either the sink rate or the compression metric at layer $\ell$. We use $\gamma = \infty$ to denote the unmodified model. The only difference between the runs is the choice of $\gamma$, so the layer-wise difference
> $\Delta Y_\ell(\gamma) := Y_\ell(\gamma) - Y_\ell(\infty)$
> can be interpreted as the causal effect of clipping at level $\gamma$ on that outcome at layer $\ell$. We estimate an average treatment effect (ATE) by averaging $\Delta Y_\ell(\gamma)$ across layers, and perform a paired test on these differences. The results are summarised in the new Figure 20 in the Appendix, which shows the effect of clipping on the norm of the massive activation, entropy, and sink rate, and in the table below :
>
> | $\gamma$ | ATE(entropy) ($\gamma$ – base) | $p$-value (entropy)     | ATE\(_\text{sink}\) ($\gamma$ – base) | $p$-value (sink)        |
> |---------:|-------------------------------:|-------------------------:|--------------------------------------:|-------------------------:|
> | 100.0    | +0.0539                        | $9.00 \times 10^{-6}$    | +0.0573                               | $5.67 \times 10^{-3}$    |
> | 10.0     | +0.1330                        | $5.74 \times 10^{-9}$    | −0.1042                               | $8.84 \times 10^{-3}$    |
> | 0.5      | +0.3002                        | $2.31 \times 10^{-10}$   | −0.6953                               | $1.44 \times 10^{-13}$   |
> | 0.1      | +0.6733                        | $1.04 \times 10^{-12}$   | −0.7266                               | $1.46 \times 10^{-13}$   |
>
> The results demonstrate strong causal effects with clear dose-response relationships, where we observe that for:
>
> **Compression (entropy):** ATE increases monotonically as clipping strengthens
> **Attention sinks:** ATE (sink rate reduction) increases with clipping
>
> Since both interventions directly manipulate the hypothesized cause (massive activation magnitude) and produce predictable changes in outcomes (compression, sinks), with effect sizes monotonically related to intervention strength, these results provide strong evidence that massive activations causally drive both phenomena, not merely correlate with them.

---

> ### Author Response · Authors · 2025-11-21
> **Reply to the Review: Part 2**
>
> >Q2: Can you trace an example showing the three stages, using linear probing to validate your theory of information flow?
>
> We would like to highlight that we have performed linear probing experiments in the final section of the paper (shown in Figure 8 and in Figure 31 in the Appendix), together with generation-focused probing, in order to validate the mix, compress, and refine theory. These experiments provide an end-to-end view of how information is organised across layers and how different stages support different computational roles. We will make this connection more explicit in the revised paper to ensure the intent of this analysis is clear.
>
> Just to reiterate the intention of these experiments: in the final section of the paper, we perform a layerwise linear probing analysis. For each layer, we embed both a training and a test set and train linear classifiers to map the embeddings to the correct answers. This directly evaluates how linearly retrievable the information is at each depth. Since linear probing evaluates the quality of the representations themselves, we treat this as an embedding-style task. In contrast, the LogitLens and TunedLens experiments evaluate the quality of the generation dynamics by measuring how well intermediate states support next-token prediction or multiple-choice answering.
>
> We find that the middle layers achieve the strongest performance on classification based on linear probing, which aligns with the compressed phase. Importantly, this peak is not limited to a single prompt but appears consistently across the entire dataset: the layers with the strongest compression are also the layers with the best embedding performance under linear probing. This suggests that embedding-style tasks benefit from representations that have been mixed and then compressed into a lower-dimensional space, which may be a natural by-product of the massive activations we study.
>
> In contrast, generation-oriented probing (via LogitLens and TunedLens) shows that performance continues to improve into the later layers. This pattern indicates that once the model has constructed a compact and mixed representation, it then transitions toward modifying and refining that representation to optimise the generation objective. To make this connection clearer, we have moved the Wikitext perplexity-evolution figure from the appendix into the main text (now Figure 8).
> We believe this distinction is important: embedding and generation tasks benefit from different stages of the information-flow pipeline, and the probing results align well with the mix, compress, and refine phases proposed in the paper.
> We thank the reviewer for this question. If we have misunderstood your intention, please let us know, and we will clarify our response accordingly.
>
> >Q3: Can you compare results on different datasets? Are the results substantially different for e.g., prose vs. math questions vs code vs scientific papers?
>
> We agree that this is an important test of generalizability.  To address this concern, we expanded our analysis to include datasets with substantially different linguistic and structural properties. In particular, we now run the same experiment on 5,000 examples each from The Stack (code), FineWeb (prose), and Hendrycks-Math (mathematical questions).
>
> **Results:**
>
> Our results confirm the phenomena generalize across diverse data modalities: massive activations emerge at the same relative depths, compression valleys coincide precisely with activation peaks, and sink rates track compression consistently across all three datasets.  We include these new results in Figures 9–11 in the Appendix, which show synchronized evolution of massive activations, compression, and sinks across all four datasets in Llama 3 8B, Pythia 6.9B, and Qwen2 7B, respectively, confirming robustness across both datasets and model families.
>
> The cross-dataset consistency demonstrates these are **architectural phenomena** driven by the interaction between transformer components, not artifacts of training distribution biases, prompt formatting conventions, domain-specific linguistic patterns, or token frequency statistics. This supports our mechanistic account: massive activations emerge from amplifying MLP dynamics (Section 4), which depend on architectural inductive biases rather than data content.
>
> We thank you once again for your careful review. We hope that our additional results and comments help to improve your opinion of our work. Given that we have carefully addressed all points raised through new experiments, additional clarifications, and a clearer presentation of our work, we would be grateful if you might consider raising your score.

---

> > ### Comment · Reviewer_qBpd · 2025-11-22
> > **Response to Authors**
> >
> > Thanks for the detailed and comprehensive rebuttal.
> >
> > > Causal analysis
> >
> > Thanks for the analysis and causality experiments. They are indeed convincing. The concerns I had about rigor in describing causality are resolved. For completeness, it might also be good to do this same experiment except ablate the MLP instead (like you did originally), but now I expect that the results will look similar.
> >
> > > Linear probing experiments
> >
> > Yes, these were there in the original draft. The experiments do show at a broad level what amount of information is linearly organized in each layer. My original intent when asking the question was that it would be really great to try to isolate what _kinds_ of information are stored where, e.g., train a linear probe to extract where the arithmetic is performed in GSM8K (where the information of the operands disappear and the answer emerges). More generally I feel that a bit more mechanistic evidence for the mix-compress-refine framework can improve the paper even more.
> >
> > > Different datasets
> >
> > Yes, these results make a great addition to the work.
> >
> >
> > Overall, since all weaknesses raised in the initial review were addressed (and in the case of Weaknesses 1 and 3, conclusively), I will raise my score.

---

> > > ### Author Response · Authors · 2025-12-01
> > > **Thank you for raising your score**
> > >
> > > We would like to thank the reviewer for the thoughtful and thorough evaluation of our work, for engaging with us throughout the process, and for raising your score from 6 to 8.

---

### Official Review · Reviewer_Wogq · 2025-10-31

**Soundness:** 3
**Presentation:** 3
**Contribution:** 4
**Rating:** 8
**Confidence:** 3

**Summary:**

This paper aligns two seemingly disjoint phenomena within transformers, attention sinks (where an attention head is dominated by attention weights for a semantically uninformative token) and compression valleys (where intermediate representations show low entropy despite high dimensionality). Both empirically and theoretically, the authors tie these phenomena to massive activations in the residual stream for the beginning of sequence token, showing that as the BOS norms increase, there is a corresponding increase in the sink rate and a decrease in entropy. The authors use these insights to present a mix-compress-refine theory of information flow, additionally tying this theory to insights regarding performance on various classes of tasks at different layers within the network.

**Strengths:**

1. The breadth of analysis is strong, with the analysis including decoder-only models of the Pythia, Llama, and Qwen families in the main paper. In particular, the analysis of Pythia 410M over training is visually illuminating of how the properties discussed emerge rapidly over training. The ablation experiments in Llama 3 8B showing that removing massive activations causally decreases attention sinks and compression are also convincing, though not general across every model.
2. The theoretical framework analyzing massive activations and their influence on compression bounds is well founded.
3. The mix-compress-refine theory of information flow that the paper presents follows reasonably naturally from the insights on experiments with massive activations. This is then made more rigorous with downstream task evaluations, evaluated at various layer representations, that show that refinement requires later layers, while embedding signals are most useful in the middle layers.

**Weaknesses:**

1. Returning to the BOS ablation experiments, looking at Figure 14, do you have any insights as to why sinks persist in Pythia 410M despite the ablation for the BOS token? Is it the same for other Pythia models, such as 6.9B?
2. Some of the figures, such as Figure 2-6, while illuminating, show only a single model. Despite many of the additional analyses showing up in the appendix, to increase robustness of the figures in the main text, it may be worth considering adding additional data to such figures when possible.
3. The ablation of the MLP contribution currently sets it to zero. A gradual decrease of the contribution from its original value to zero would be very insightful. Do you see a gradual, dose-response effect in terms of the changes to entropy and sink rate as a result of a gradual ablation?

Overall, despite these minor weaknesses, the breadth of analysis of the paper combined with the novel results tying together two seemingly disparate phenomena, as well as the mix-compress-refine framework, motivate my score of 8.

**Questions:**

1. See weaknesses.
2. Additionally for Figure 14, while the general trend of ablations decreasing sink rate and increasing entropy is supported, for the ablated entropy metric for Llama 3 8B, is there any intuition as to why there is a sudden increase in layer 15, and for Qwen 2 7B, is there any intuition for the increased sink rate in layers 4-6?

---

> ### Author Response · Authors · 2025-11-21
> **Reply to the Review**
>
> We thank the reviewer for the careful reading of the paper and the constructive suggestions, as well as the positive evaluation! We found the comments on the Pythia ablations, the request for stronger multi-model evidence in the main text, and the idea of gradual (dose-response) interventions particularly helpful.
>
> > Returning to the BOS ablation experiments, looking at Figure 14, do you have any insights as to why sinks persist in Pythia 410M despite the ablation for the BOS token? Is it the same for other Pythia models, such as 6.9B?
>
> Thank you for highlighting this. Reviewer FFpb raised the same concern (Q1, Weakness 3). Following both reviewers' suggestions, we conducted a systematic re-examination of all ablation experiments.
>
> In the original version, we were only ablating the BoS-directed MLP contribution at layers 5–7. However, Pythia 410M already develops a non-trivial BoS norm spike at the very first MLP layer. By leaving that early spike untouched, the intervention removed the later, larger massive activation and the associated compression, but still preserved a smaller discrepancy between the BoS norm and the norm of the rest of the tokens. This residual gap was insufficient to generate measurable compression, but sufficient to bias softmax attention patterns toward the BOS token. This sensitivity aligns with the reviewer's observation and confirms that compression requires larger norm disparities than attention sinks.
>
> **Corrected experiment:** We now ablate the MLP contribution starting from the earlier layer that creates the initial norm spike (the first one). Under this updated intervention, sink rates drop from ~100% in the middle layers to ~10% and now track the disappearance of compression, in line with our original picture. We have updated the corresponding figure (now Figure 18) to reflect this.  We are grateful for this question, which helped us identify and correct an important detail in the original ablation design.
>
> > Some of the figures, such as Figure 2-6, while illuminating, show only a single model. Despite many of the additional analyses showing up in the appendix, to increase robustness of the figures in the main text, it may be worth considering adding additional data to such figures when possible.
>
> We agree with the reviewer that showing only a single model in the main text underplays the robustness of the phenomena. In response to this, we have moved the norm equalization plots for Pythia 410M, Llama3 8B, and Qwen2 8B, as well as the evolution of the entropy, sink rate, and massive activation norm during training for Pythia 6.9B, to the main text.
>
> > The ablation of the MLP contribution currently sets it to zero. A gradual decrease of the contribution from its original value to zero would be very insightful. Do you see a gradual, dose-response effect in terms of the changes to entropy and sink rate as a result of a gradual ablation?
>
> We agree that a graded intervention is more informative than a single on/off ablation. Motivated by your suggestion and similar comments from reviewers FFpb and qBpd, we implemented a dose-response style intervention directly on the massive activation channel. Rather than only setting the BoS MLP output to zero, we progressively clip the massive activation channel to different thresholds $\gamma$ at each layer and then measure the resulting entropy and sink rate. This gives us a family of models indexed by $\gamma$, ranging from the original model ($\gamma = \infty$) to strongly clipped versions.
>
> The results, (Figure 20) show clear monotonic relationships:
>
> - Entropy increases from 0 to 0.8 as $\gamma$ decreases (stronger clipping $\implies$ less compression)
> - Sink rate decreases from 100% to 0% as $\gamma$ decreases (weaker BOS norm $\implies$ fewer sinks)
>
> Notably, compression responds more smoothly to norm reduction while sink rates show higher variance, consistent with the sensitivity observed in the Pythia 410M ablation (Question 1).
> Additionally, we quantify these effects using an average treatment effect (ATE) analysis. For each clipping level $\gamma$, we compare the model’s entropy and sink rate to the unmodified model and average the resulting differences. This provides a formal estimate of how reducing the massive activation norm causally affects compression and sinks. The results show strong, statistically significant dose–response patterns. For a more detailed explanation of the ATE setup and results, we refer the reviewer to our response to reviewer qBpd.

---

> ### Author Response · Authors · 2025-11-21
> **Reply to the Review: Part 2**
>
> > Additionally for Figure 14, while the general trend of ablations decreasing sink rate and increasing entropy is supported, for the ablated entropy metric for Llama 3 8B, is there any intuition as to why there is a sudden increase in layer 15, and for Qwen 2 7B, is there any intuition for the increased sink rate in layers 4-6?
>
> We appreciate the reviewer drawing attention to these local irregularities. We interpret them as follows:
>
> **Llama 3 8B entropy bump (layer 15):** After the ablation, a small local increase in entropy occurs because token norms become particularly similar, which flattens the singular value distribution.
>
> **Qwen2 7B sink persistence (layers 4-6):** We interpret this behaviour as another instance of the greater sensitivity of sink formation to small norm imbalances. Even after ablation, a few early layers in Qwen2 7B retain slight discrepancies between the BoS norm and the average token norm. Although these differences are too small to produce measurable compression, they can still bias the softmax strongly enough to generate a sink pattern. This is now shown explicitly in Figure 17.
>
> We note, however, that these artifacts do not alter the primary finding: massive activation removal consistently reduces compression and sinks across the bulk of middle layers.

---

> > ### Comment · Reviewer_Wogq · 2025-11-26
> > **Reply to authors**
> >
> > I would like to thank the authors for their interesting new experiments, as well as for their answers to my questions. I retain my positive score and view of the paper.

---

> > > ### Author Response · Authors · 2025-12-01
> > > **Thank you for your positive evaluation**
> > >
> > > We thank the reviewer for their engagement and positive evaluation of our work!

---

### Official Review · Reviewer_8eqz · 2025-11-01

**Soundness:** 1
**Presentation:** 3
**Contribution:** 2
**Rating:** 4
**Confidence:** 4

**Summary:**

This paper argues that attention sinks and compression valleys in LLMs are caused by the same underlying mechanism—massive activations in the residual stream, often on the BOS token. The authors show empirically that these phenomena always co-occur in the middle layers across models (410M–120B). They prove that when one token’s activation norm dominates, it forces a dominant singular value in the representation matrix, causing low entropy and thus compression. They confirm the causality through ablations. Based on this, they propose a Mix–Compress–Refine theory of information flow: early layers mix information broadly, middle layers compress representations and halt attention mixing, and late layers re-expand and refine outputs. The paper also connects these phases to task performance—embedding tasks perform best during compression, while generation benefits from full refinement. Overall, the work unifies two previously separate observations and provides a clear mechanistic view of how transformers structure computation across depth.

**Strengths:**

The three-phrase part seems to be interesting. They propose and try to demonstrate a possible three-phase structure. In early layers, attention entropy is high and attention patterns are broadly mixed (Fig. 6 left). In middle layers, mixing decreases and representations become more compressed (Fig. 6 middle). In later layers, attention shifts to localized positional patterns and token norms become more balanced (Fig. 6 right). Task-level results further align with this trend: embedding performance tends to peak around the mid-layer compression phase, while generation tasks continue improving toward the final layers (Figs. 7, 23–27).

**Weaknesses:**

The paper has two parts—the relationship between attention sinks and compression valleys, and the three-phase structure. However, I do not find either of them satisfying enough.

1. The connection between attention sinks and compression valleys is quite trivial. It is well known that attention sinks and massive activations occur simultaneously. Then a massive activation leads to a dominant largest singular value in the activation matrix. The causal ablations that remove early-layer MLPs are not new either.

2. It was observed previously that early layers have more uniform (or mixed) attention patterns, middle layers have attention sinks, and late layers show special functionality. The paper develops these observations into the three-phase structure too strongly, without providing stronger evidence. Figures 6 and 7 simply reproduce previous experiments and cannot support stronger claims.

**Questions:**

Please see weakness

---

> ### Author Response · Authors · 2025-11-21
> **Reply to the Review**
>
> We thank the reviewer for taking the time to engage closely with our work and for offering clear and thoughtful feedback. We hope the revisions address your concerns. If any issues remain unclear or you feel further clarification is warranted, we would be very glad to elaborate.
>
> > The connection between attention sinks and compression valleys is quite trivial. It is well known that attention sinks and massive activations occur simultaneously. Then a massive activation leads to a dominant largest singular value in the activation matrix. The causal ablations that remove early-layer MLPs are not new either.
>
> We thank the reviewer for this comment. While prior work has documented a connection between attention sinks and massive activations, there is no in-depth, layerwise analysis of these phenomena. For example, Sun et al. (2024) suggest that, in some models, the emergence of massive activations contributes to the appearance of sinks, but this is neither quantified nor evaluated per layer. On the other hand, Skean et al. (2025) studied compression valleys, yet the underlying mechanism that produces these valleys has remained unclear, and attention sinks or massive activations are not discussed in that context.
>
> Our work aims to fill this gap by making these relationships explicit and mechanistic. We show that massive activations on the BOS token directly induce spiked singular value spectra, and we provide formal bounds linking the magnitude and alignment of this activation to anisotropy and matrix-based entropy. These bounds are tight precisely in the compressed middle layers, and our empirical results confirm that massive activations largely determine the observed singular value structure. This helps clarify why compression valleys arise in LLMs and challenges the common narrative that middle layers encode “rich geometric representations” without a concrete mechanism.
>
> In terms of attention sinks, our analysis extends prior observations by demonstrating that the characteristic rise and later disappearance of massive activations naturally produces the three-phase structure we identify. The massive activation not only contributes to sinks, consistent with the over-mixing perspective of Barbero et al. (2025), but also compresses representations into low-dimensional subspaces before the late refinement phase.
>
> This unification has two important implications:
> 1. Researchers using singular-value diagnostics (entropy, anisotropy) to assess "geometric richness" should recognize these metrics may reflect sink-token artifacts rather than content-token structure when massive activations are present.
> 2.  The co-occurrence of sinks and compression is not coincidental, both are forced by the same massive activation structure, explaining why interventions on one affect the other (see new Figure 20).
>
> Regarding the causal ablations, several reviewers commented on these experiments, and we have substantially revised and extended them in response. To this end, we point the reviewer to the answers provided to Reviewers FFpb and Wogq for explanation on additional experiments on progressive norm-reduction of the massive activation, and an ATE-based analysis across layers. We hope the revised experiments and clearer framing address the reviewer’s concerns.

---

> ### Author Response · Authors · 2025-11-21
> **Reply to the Review: Part 2**
>
> > It was observed previously that early layers have more uniform (or mixed) attention patterns, middle layers have attention sinks, and late layers show special functionality. The paper develops these observations into the three-phase structure too strongly, without providing stronger evidence. Figures 6 and 7 simply reproduce previous experiments and cannot support stronger claims.
>
>
> We appreciate this comment. We would like to clarify that our aim in this section is not to claim novelty for the individual tools we apply, but to connect the three phases to concrete functional roles. More broadly, our contribution in this part of the paper is fourfold:
>  (i)  a mechanistic account of why the transitions occur, based on the emergence and disappearance of massive activations,
>  (ii)  quantitative markers of the phase boundaries via entropy, sink rate, and norm-based metrics,
>  (iii)  causal evidence, since targeted ablations shift or remove the phase transitions exactly as predicted, and
>  (iv)  functional validation, showing that downstream task performance aligns with these phases.
>
> With that intention, we use linear probing and LogitLens as complementary tools to connect the theoretical picture to downstream behaviour. First, we confirm that intermediate layers perform best on embedding-style tasks. Second, we add a layerwise linear probing experiment on the same datasets for both embedding- and generation-oriented objectives. This side-by-side comparison shows that the layers that are best for embedding differ from those that are best for generation, and that these differences align closely with the three phases we describe. In particular, embedding tasks peak in the compressed middle phase, while generation performance improves into the late refinement phase. This task-dependent behaviour provides further support that the phases correspond to distinct computational purposes. To make this distinction clearer, we have also moved the perplexity-evolution figure from the Appendix into the main text (Fig 8).
>
> Regarding Figures 6 and 7 (now 6 and 8) specifically, our intention is to link the more abstract mechanistic observations to concrete downstream performance. These visualisations help illustrate how the phases we identify connect to different functional behaviours, but they are not the primary evidence for the phase structure. The stronger support comes from the earlier metrics and analyses: the coordinated evolution of massive activations, sinks, and entropy; the theoretical bounds that become tight in the compressed layers; the targeted ablations; and the systematic downstream evaluations across tasks.

---

### Official Review · Reviewer_FFpb · 2025-11-12

**Soundness:** 2
**Presentation:** 3
**Contribution:** 2
**Rating:** 4
**Confidence:** 3

**Summary:**

This paper investigates the relationship between attention sinks and compression valleys in LLMs, proposing that both phenomena arise from massive activations in the residual stream.


The authors prove that "massive activations" necessarily produce representational compression through singular value dominance.


Through experiments across diff. models they validate that when the beginning-of-sequence (BOS) token develops extreme activation norms, both compression valleys and attention sinks emerge simultaneously.

The paper provides new insights, but its theoretical contributions might not be strong. In my opinion,  most of the technicalities are based on well-known linear algebra.

**Strengths:**

- The paper is well written with good contextualization relative to prior work,
and comprehensive appendix with additional validation.


- While the proof technique is straightforward (using Rayleigh quotient characterization), this is the *first* work to formally prove that massive activations $M = ||x_0||^2 $ necessarily induce spectral dominance:  $\sigma_1^2 \geq M + \alpha R$.

-   The derived bounds on dominance  $\frac{\sigma_1^2}{\sum_{j \geq 2} \sigma_j^2} \geq \frac{c+\alpha}{1-\alpha}$,  anisotropy $p_1 \geq \frac{c+\alpha}{c+1}$,  and entropy are mathematically sound.

-   Figure 3 shows that in middle layers (where massive activations emerge), the theoretical bounds closely match empirical values, validating that massive activations are the dominant mechanism shaping representation geometry. Targeted MLP ablations (Figures 4 and 14) across multiple models provide causal evidence.

**Weaknesses:**

- The theorem relies on well-known linear algebra and, in my opinion, lacks strong theoretical novelty.

- The bound depends on $\alpha = \frac{1}{R}\sum_{i \neq 0} \|x_i\|^2 \cos^2\theta_i$, but Figure 3 suggests $\alpha R \ll M$ in practice.
More analysis of what determines $\alpha$ would strengthen the work.

- Pythia 410M ablations show that compression is removed without eliminating sinks, suggesting that sinks may have multiple emergence mechanisms.
Since ablations only target MLP contributions, direct manipulations of massive activations (e.g., through regularization or initialization) would provide stronger causal evidence.

- Lines 200–206: The proof sketch is potentially misleading, as it may imply circular reasoning.
You may clarify that the Rayleigh characterization provides a lower bound when choosing $v = x_0 / \|x_0\|$.

- Lines 242–247: This contradicts the claim that massive activations jointly cause sinks and compression.
You may either analyze why Pythia 410M behaves differently or restate the claim as a "common but not universal" mechanism.

- Lines 185–187: I don’t think $r = 0.58 \pm 0.25$ indicates a strong correlation; it appears to be only moderate.

- Lines 206, Figure 3: Although $\alpha R \ll M$, $\alpha$ appears crucial, yet there is no analysis of its origin or importance.

- Lines 185–186, 907–909: No p-values are reported, the sample size ($n = 6$) is small, and there is no correction for multiple testing.
What are the $p$-values and sample size used?

- Lines 421–424: Some parameters might need to be reported, such as the optimizer, learning rate, batch size, number of epochs, and random seed.

**Questions:**

Q0. Please see Weaknesses

Q1.  Why do Pythia 410M ablations remove compression but not sinks, and does this indicate that sinks can emerge through multiple independent mechanisms?

Q2.  What determines $\alpha = \frac{1}{R}\sum_{i \neq 0} \|x_i\|^2 \cos^2\theta_i$, and since Figure 3 shows $\alpha R \ll M$, is the alignment term theoretically necessary or practically negligible?

Q3. Have you tried inducing massive activations artificially (e.g., via initialization or regularization) to test whether sinks or compression follow?

Q4. Can phase transition points be predicted from architectural parameters such as depth, width, or number of heads rather than observed post hoc?

Q5. Do these phenomena appear in encoder-only models or models with alternative positional encodings, and how do these differences affect the emergence of massive activations and phase structure?

---

> ### Author Response · Authors · 2025-11-21
> **Response to the Review**
>
> We thank the reviewer for the thoughtful and constructive feedback. Several of your comments led directly to substantive improvements in the manuscript, especially regarding the alignment term, the ablation analysis, and the statistical reporting. We are grateful for your careful reading, and we hope that the clarifications and new results in the revised version address your concerns and lead you to reconsider your overall evaluation.
>
> > The theorem relies on well-known linear algebra and, in my opinion, lacks strong theoretical novelty.
>
> We agree that the proof techniques themselves (Rayleigh quotient and singular-value bounds) are standard linear algebra, and we do not claim mathematical novelty at the level of new theorems in matrix analysis. Our goal is different: to give a simple, mechanistic explanation for the well-documented “middle-layer phenomena” in LLMs. In particular, the anisotropy/compression valleys reported by Razzhigaev et al. (2024) and Skean et al. (2025). These works carefully measure that intermediate layers are highly compressed and empirically useful, but they do not explain why such a compression valley appears, nor what concrete circuit-level mechanism produces the observed singular-value spectrum.
>
> Our contribution is to show that once you choose to quantify compression via singular-value–based metrics (entropy, anisotropy, etc.), **the presence of a massive activation on a single token is already enough structure to almost fully determine those metrics**. In other words, given a representation matrix where one row has a much larger norm than the rest, the induced spike in the singular-value distribution, and hence the drop in entropy and rise in anisotropy, is mathematically forced. The theorem formalizes this and gives quantitative bounds that are (i) tight precisely in the regime where compression valleys occur in real models, and (ii) predictive across architectures and scales. This is what lets us argue that massive activations are not just a correlate but the dominant cause of the mid-layer compression seen in prior work, and to connect this directly to attention sinks within our Mix–Compress–Refine framework.
>
> We actually see the simplicity of the argument as a feature rather than a limitation: because the mechanism is transparent, it both (a) provides a clear mechanistic story linking massive activations, sinks, and compression valleys, and (b) highlights a caveat for the community when interpreting singular-value–based diagnostics in LLMs. In models where such massive activations exist, these metrics may be dominated by sink tokens rather than by the geometry of content-token representations.
>
> To reflect this better, we are happy to clarify in the paper that the novelty lies not in new linear-algebraic techniques, but in this mechanistic link and in the observation that singular-value–based compression metrics are largely determined by the massive-activation structure that modern LLMs naturally develop.
>
> > Importance of the alignment term $\alpha$
>
> Thank you for this very helpful comment and for drawing our attention to the role of the alignment term
> $\alpha$. We agree that our original discussion did not make this sufficiently clear.
>
> Mathematically, we introduced $\alpha$ to capture the average alignment of non-BOS tokens with the BOS direction. We introduced this term for mathematical completeness. However, as the reviewer correctly observes, in the regimes where massive activations appear, we empirically have $\alpha R \ll M$. In practice, the bounds and the observed compression are therefore dominated by the BOS norm $M$, with $\alpha$ providing only a small correction. In the revision we now make this explicit in the text and updated the corresponding plots (Figs. 15 and 16) that separate the bounds with and without the alignment term, where **we observe that precisely in the middle layers (where compression is strongest and the bounds become tight) the $M$ term dominates**.
>
> We are grateful for the suggestion: it led us to clarify the origin and practical importance of $\alpha$, and to add empirical evidence that the dominant driver of compression is the BOS norm gap, with alignment playing a secondary role in the models we study. We now clarify in Section 3.2 (l. 235-237) that $\alpha$ captures alignment but $\alpha R$ is empirically negligible compared to $M$ in the compression regime. We have also added a brief explanation of the weaker bounds obtained without the $\alpha$ term in the appendix, corresponding to Figs. 15 and 16. We hope that this addresses your concern.

---

> ### Author Response · Authors · 2025-11-21
> **Response to the Review: Part 2**
>
> > Why do Pythia 410M ablations remove compression but not sinks, and does this indicate that sinks can emerge through multiple independent mechanisms?
>
> We thank the reviewer for these comments and for pointing us to the discrepancy observed in the Pythia-410M ablations. Following this suggestion, we performed a more careful analysis to understand why compression was removed while sinks persisted.
>
> We discovered that in our original setup, we removed the MLP contribution at layers 5-7, inadvertently leaving an earlier BOS-norm spike from the first layer. This weak spike was insufficient for compression but still triggered sinks due to softmax sensitivity to even moderate norm differences. After ablating layer 1 as well, **both sinks and compression now disappear together** (new Figure 18), confirming the unified mechanism. Furthermore, we have added a new analysis (Figure 19 in the updated paper in the Appendix), which identifies the specific MLP layers that create and dissolve massive activations, confirming their formation is localized to a few critical layers rather than distributed across the network. This fully resolves the contradiction noted previously in Lines 242-247 and confirms massive activations are the common cause.
>
> Furthermore, following your suggestion and those of reviewers Wogq and qBpd, we performed an additional ablation in which we progressively reduced the norm of the massive activation itself. This allows us to directly examine how weakening the massive-activation pathway affects the emergence of sinks and compression. The updated results, shown in the new Figure 20, demonstrate that as the magnitude of the massive activation is gradually diminished, both sinks and compression consistently disappear. We also refer the reviewer to our response to reviewer qBpd, where we present a more extensive causal analysis of this effect.
>
> > The proof sketch is potentially misleading, as it may imply circular reasoning.
>
> Thank you for this comment. We have removed the proof sketch and now instead directly point to the Appendix for a full proof.
>
> > Correlation of massive activation to sinks
>
> We thank the reviewer for raising these points. The concern about the magnitude of the reported correlation ($r = 0.58  \pm 0.25$) is valid. After revisiting the analysis, we found that Llama-3-8B is an extreme outlier for this particular computation: almost all its layers exhibit very large BOS activations, which means that first-order differences are nearly flat, making the per-layer correlation ill-defined. This artificially suppresses the correlation magnitude and inflates its variance. We will now explicitly note this in the text.
>
> To show this, we report below the correlations for each model separately. Furthermore, we provide p-values and Bonferroni correction for each one.
>
>
> | Model              | Correlation BOS Norm and Entropy | p-value   | p-value (Bonf.) |
> |------------------------|-----------------------------------|-----------|-----------------|
> | Qwen2-7b       | -0.987145                         | 0.00e+00  | 0.00e+00        |
> | pythia-6.9b     | -0.984760                         | 0.00e+00  | 0.00e+00        |
> | pythia-410m     | -0.841406                         | 3.34e-06  | 4.01e-05        |
> | Meta-Llama-3-8B | -0.797692                         | 1.30e-07  | 1.56e-06        |
> | bloom-1b7      | -0.746520                         | 3.73e-04  | 4.47e-03        |
> | gemma-7b        | -0.452089                         | 7.87e-02  | 9.45e-01        |
>
>
> | Model             | Correlation BOS Norm and Sink | p-value   | p-value (Bonf.) |
> |------------------------|-------------------------------|-----------|-----------------|
> | Qwen2-7b      | 0.806043                      | 2.26e-07  | 2.71e-06        |
> | pythia-6.9b    | 0.538409                      | 6.64e-03  | 7.97e-02        |
> | pythia-410m      | 0.600424                      | 5.12e-03  | 6.15e-02        |
> | Meta-Llama-3-8B  | 0.018071                      | 9.25e-01  | 1.00e+00        |
> | bloom-1b7        | 0.737464                      | 4.78e-04  | 5.74e-03        |
> | gemma-7b       | 0.523146                      | 3.76e-02  | 4.51e-01        |
>
> > Some parameters might need to be reported, such as the optimizer, learning rate, batch size, number of epochs, and random seed.
>
> We thank the reviewer for pointing this out. The only experiments requiring training details are the linear probing experiments (embedding tasks), and several of these parameters are already reported at the end of Appendix B (lines 1298-1300). We have also added some missing parameters at the reviewer’s suggestion.

---

> ### Author Response · Authors · 2025-11-21
> **Response to the Review: Part 3**
>
> > Do these phenomena appear in encoder-only models or models with alternative positional encodings, and how do these differences affect the emergence of massive activations and phase structure?
>
> We thank the reviewer for this question. Our empirical study focuses on decoder-only models, including both RoPE and non-RoPE variants. We find:
> 1. Massive activations and compression appear in both RoPE and non-RoPE decoder models
> 2. *Phase 3 differs*: RoPE models show sharp positional heads (identity/previous-token) while non-RoPE models (Bloom) show partial return to diffuse mixing, consistent with Barbero et al. 2025b showing RoPE enables frequency-selective patterns
> 3. *Encoder-only models*: Skean et al. 2025 report no compression valleys in BERT (~0.9 across layers for the normalized entropy). We hypothesize this relates to bidirectional attention enabling more uniform information flow, but detailed analysis is needed. Our theoretical bounds (Theorem 1) apply regardless of architecture, so if encoder models avoid compression valleys, then they also avoid massive activations.
>
> We thank you once again for your valuable comments and for carefully reviewing our paper. We hope that our replies and improvements to the paper have addressed your concerns. We are very happy to reply to any further points! Further, given that we have invested significant effort in addressing each of your concerns through extensive new experiments and theoretical analysis, and the substantial improvements made to address all raised issues while maintaining the core strengths you recognized, we would greatly appreciate your consideration for raising the score.

---

### Author Response · Authors · 2025-11-21
**General Response to all Reviewers**

We thank the reviewers for their timely and thoughtful feedback. We have made several targeted improvements in direct response to the comments, and updated or added figures and explanations are highlighted in blue in the revised manuscript.

Summary of key updates:

**Sharper theoretical clarification.** Following Reviewer FFpb, we updated figures separating the effects of $M$ and $\alpha$ in the derived bounds and clarified why the BOS norm dominates in practice.

**Corrected and strengthened ablations.** As suggested by Reviewers FFpb and Wogq, we fixed the Pythia-410M ablation by also removing the early MLP-induced norm spike. In the updated results, sinks and compression now disappear together (new Fig 18).

**Dose-response and causal analysis.** Motivated by Reviewers FFpb, qBpd, and Wogq, we added graded clipping interventions on the massive activation and an ATE analysis. These show clear dose-response effects on entropy and sink rate (new Fig. 20).

**Broader robustness and multi-model coverage.** Per Reviewer Wogq's request, we moved several cross-model results (Pythia 6.9B, Llama3 8B, Qwen2 7B) from the appendix into the main text.

**Cross-dataset generalization.** In response to Reviewer qBpd, we added experiments on code, prose, and math datasets (Figs. 9–11), showing consistent activation/compression/sink behaviour across domains.

We are grateful for the reviewers’ suggestions, which have helped us greatly improve clarity and robustness. We hope these revisions address the raised concerns and will be taken into account in the final evaluation.

---

### Author Response · Authors · 2025-12-01
**Overall Comment to the New AC**

Dear AC,

Thank you for handling our paper and for overseeing the discussion. We wanted to briefly highlight the extent of the changes made in response to the reviews. In the revised version (changes in blue), we (i) sharpened the theoretical exposition by separating the roles of the BOS norm and the alignment term, (ii) corrected and extended the BOS/MLP ablations (including for Pythia-410M), (iii) added graded clipping and ATE-style analyses that provide clearer causal evidence linking massive activations, compression, and sinks, and (iv) broadened the empirical coverage with additional models and datasets (prose, code, and math).

In spite of the recent decisions made by ICLR 2026, we remain very grateful to the reviewers for their time and engagement. During the discussion,  reviewer Wogq **maintained their positive score of 8**, while reviewer qBpd explicitly stated that their main concerns had been addressed by the new experiments and clarifications and **raised their score from 6 to 8**. Sadly, we did not receive further feedback from the remaining reviewers before the freeze, but we sincerely appreciate their initial comments, which also shaped the revision. We believe that the substantial new analyses and experiments directly address most of their concerns, and we hope this is clear from our detailed replies and is taken into account in your decision.

We are disappointed not to have been able to continue the rebuttal process, but we thank all reviewers and the AC for their efforts and for considering our work under these unusual circumstances.

Best,

Authors

---

### Meta-Review · Area_Chair_6Qzu · 2026-01-01

**Summary:**

This paper makes a clear, mostly well-supported case that attention sinks and compression valleys are linked through the same underlying mechanism: massive activations (often on the BOS token) in the residual stream. Reviewers generally agreed the writing and empirical coverage are strong, and that the work is useful as a unifying, mechanistic account across multiple model families and scales. The main disagreements were about (i) how novel the theoretical contribution is (given it leans on standard linear algebra), and (ii) whether the "Mix–Compress–Refine" framing is sufficiently evidenced beyond prior qualitative observations.

Overall, the rebuttal and revision meaningfully strengthened the causal story and robustness: the authors corrected and extended key ablations (including addressing the Pythia-410M discrepancy), added graded/dose-response clipping interventions and an ATE-style causal analysis, improved statistical reporting, and broadened datasets beyond the original focus.

**Reviewer Concerns:**

### Concerns substantially addressed in the rebuttal ###

- Multiple reviewers asked for more direct interventions on massive activations. The revision adds graded clipping and an ATE-style analysis showing a clear dose-response effect on both entropy/compression and sink rate, which directly targets the hypothesized cause.

- Pythia-410M ablation contradiction (compression removed but sinks remained): The authors report this was due to an unablated early BOS norm spike and provide a corrected ablation where sinks and compression now disappear together, resolving the key inconsistency.

- Concerns that some figures were single-model are partially mitigated by moving additional cross-model results into the main paper and adding broader dataset coverage (code/prose/math).

- Reviewer FFpb requested clearer handling of the alignment term and stronger reporting (p-values, sample sizes, training details where relevant). The rebuttal explicitly separates the BOS-norm and alignment effects and adds correlation breakdowns with p-values/Bonferroni correction, plus missing experimental details for the probing setup.

### Concerns partially outstanding (but not decision-blocking) ###

- One reviewer views the theory as relying on standard linear algebra and questions novelty. The rebuttal reframes novelty as mechanistic linkage and predictive tightness in the regime of interest, which is reasonable, but it does not fully change the underlying novelty judgment.

- A reviewer felt the phase framing of "Mix–Compress–Refine" overstates what the current experiments can support and largely echoes prior observations. The authors add more task-level probing evidence and argue for quantitative markers and ablation-shifted boundaries, which improves the case, though some readers may still see the framing as a high-level interpretation rather than a strict claim.

**Reviewer Scores:**

- Reviewer Wogq: 8 to 8. This reviewer was positive throughout and would likely maintain their strong score after discussion.

- Reviewer qBpd: 6 to 8. The main requests for stronger causal evidence and broader validation were addressed, making a clear score increase as stated by reviewer.

- Reviewer FFpb: 4 probably to 6. Initial concerns about inconsistencies, missing details, and weak causal support were largely resolved, although some reservations about novelty may remain.

- Reviewer 8eqz: 4 probably to 6. While still cautious about the conceptual framing, this reviewer’s core technical concerns were addressed well enough to move the paper into borderline-positive territory.

---

### Decision · Program_Chairs · 2026-01-26

Accept (Poster)